# ParalESN: Enabling parallel information processing in Reservoir Computing

**Matteo Pinna** [* 1]  **Giacomo Lagomarsini** [* 1]  **Andrea Ceni** [1]  **Claudio Gallicchio** [1]

## Abstract

Reservoir Computing (RC) has established itself as an efficient paradigm for temporal processing. However, its scalability remains severely constrained by the need to process temporal data sequentially and the prohibitive memory footprint of high-dimensional reservoirs. To address these limitations, we revisit RC through the lens of structured operators and state space modeling, introducing Parallel Echo State Network (ParalESN). Leveraging diagonal linear recurrence in the complex domain, ParalESN enables parallel processing of temporal data and the construction of efficient, high-dimensional reservoirs. A thorough theoretical analysis demonstrates that the Echo State Property and the universality guarantees of traditional Echo State Networks are preserved, while also admitting an equivalent representation of arbitrary linear reservoirs in the complex diagonal form. Empirically, ParalESN achieves competitive predictive accuracy with traditional RC and with fully trainable sequence models, while delivering computational savings by orders of magnitude. Overall, ParalESN offers a scalable and principled pathway for integrating RC within the deep learning landscape[2].

## 1. Introduction

Reservoir Computing (RC) has emerged as a simple yet powerful paradigm for harnessing the rich dynamics of recurrent systems for learning and prediction (Nakajima & Fischer, 2021; Lukoševičius & Jaeger, 2009). By fixing a non-linear recurrent reservoir and training only a linear readout, RC offers favorable training efficiency, strong performance on temporal processing tasks, and intriguing connections to both neuroscience and dynamical systems theory. These properties have led to its success in a wide range of domains, from speech recognition to chaotic signal prediction. Despite these strengths, RC faces the same problem as traditional, fully-trainable Recurrent Neural Networks (RNNs): the input signal has to be processed sequentially, which makes training slow and not parallelizable. Additionally, scaling reservoirs to truly high dimensions is often impractical due to the prohibitive memory footprint of dense matrices. In an attempt to improve the efficiency of RC, structured operators have been investigated (Rodan & Tino, 2011; Dong et al., 2020; D'Inverno & Dong, 2025). A particularly active research area involves hardware implementations of RC (Gallicchio & Soriano, 2025). An important theoretical insight is that even *linear reservoirs*, provided that the readout is expressive enough, are universal approximators in the class of fading memory filters (Grigoryeva & Ortega, 2018a;b), thus being able to represent arbitrary input-output dynamics well.

In this work, we address the limitations of traditional RC by revisiting reservoir construction through the lens of structured operators. We introduce *Parallel Echo State Networks (ParalESN)*, a novel class of efficient untrained RNNs based on diagonal linear recurrence in the complex domain, where the recurrence can be parallelized via associative scan. The proposed model is highlighted in Fig. 1. By combining the dynamical richness of RC with the linear recurrence from Linear Recurrent Unit (LRU) (Orvieto et al., 2023), we bridge the gap between dynamical systems-inspired learning and contemporary large-scale sequence modeling. The remainder of this work is organized as follows. Section 2 discusses related works. Section 3 introduces ParalESN. Section 4 details our theoretical analysis. Section 5 presents the experiments. Section 6 concludes the paper. The Appendix is dedicated to computational complexity analyses, mathematical proofs and definitions, experimental methodology, and additional experiments. See Appendix A for a table of contents.

## 2. Related Works

**Reservoir Computing.** Reservoir Computing (RC) (Verstraeten et al., 2007; Nakajima & Fischer, 2021) is a pop-

---

*Equal contribution. [1]Department of Computer Science, University of Pisa, Pisa, Italy. [2]Code available at: https://github.com/nennomp/paralesn. Correspondence to: Matteo Pinna <matteo.pinna@di.unipi.it>, Giacomo Lagomarsini <giacomo.lagomarsini@phd.unipi.it>.

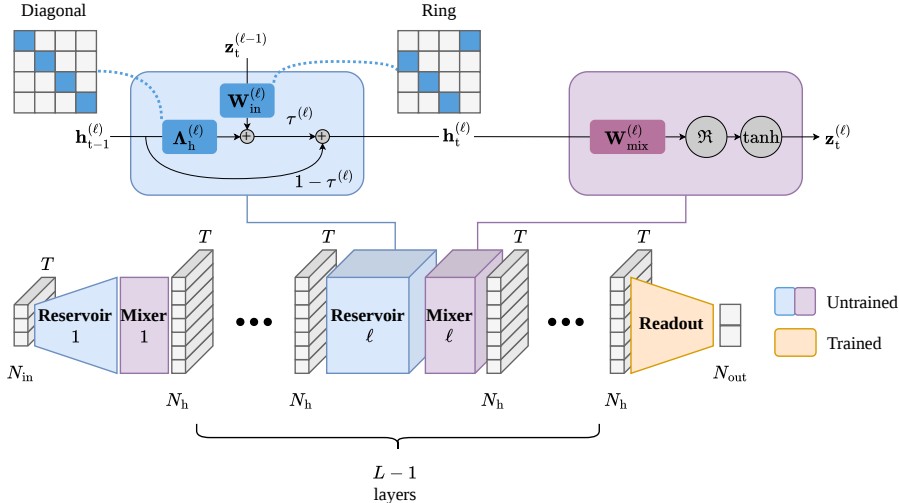

*Figure 1.* Architectural organization of the proposed ParalESN. The model may have multiple blocks consisting of two components: (i) a linear reservoir and (ii) a non-linear mixing layer. The first block processes the external input as in a traditional shallow architecture. Subsequent blocks process the output of the previous block's mixing layer, $\mathbf{z}_t^{(\ell-1)}$. The structure of the reservoir (in blue) includes a diagonal, complex-valued transition matrix $\mathbf{\Lambda}_h^{(\ell)}$ and a ring, complex-valued input weight matrix $\mathbf{W}_{in}^{(\ell)}$. The main branch and the temporal residual connections are scaled by a positive coefficient $\tau$ and $1 - \tau$, respectively. The recurrence can be easily parallelized via associative scan. The mixing layer (in purple) is used to introduce non-linearity in the model's dynamics and to mix the components of the reservoir states at each time step. The readout (in orange) is the only trainable component. See Section 3 for details.

ular framework for the design of efficient untrained Recurrent Neural Networks (RNNs), developed to address the instabilities of training RNNs (Bengio et al., 1994; Glorot & Bengio, 2010; Pascanu et al., 2013). An RC model consists of two main components: (i) a high-dimensional recurrent layer, called *reservoir*, that is randomly initialized and then left untrained, and (ii) a trainable readout layer, that may be trained via lightweight closed-form solutions. Therefore, by design, RC models bypass backpropagation and related vanishing/exploding gradients, and can be trained exceptionally fast in a single forward pass.

Echo State Networks (ESNs) (Jaeger et al., 2007) established themselves as one of the most successful instances of RC models. ESNs dynamics can be defined as follows:

$$\mathbf{h}_t = (1 - \tau)\mathbf{h}_{t\text{-}1} + \tau\,\sigma(\mathbf{W}_h\mathbf{h}_{t\text{-}1} + \mathbf{W}_{in}\mathbf{x}_t + \mathbf{b}). \quad (1)$$

where $\mathbf{h}_t \in \mathbb{R}^{N_h}$ and $\mathbf{x}_t \in \mathbb{R}^{N_{in}}$ are, respectively, the state and the external input at time step $t$. The transition matrix is denoted as $\mathbf{W}_h \in \mathbb{R}^{N_h \times N_h}$, the input weight matrix is denoted as $\mathbf{W}_{in} \in \mathbb{R}^{N_h \times N_{in}}$, $\mathbf{b} \in \mathbb{R}^{N_h}$ denotes the bias vector, $\sigma$ denotes an element-wise applied non-linearity, and $\tau \in (0, 1]$ denotes the leaky rate hyperparameter. The entries of the matrix $\mathbf{W}_{in}$ and $\mathbf{b}$ are generally sampled randomly from a uniform distribution (Gallicchio et al., 2017; Ceni & Gallicchio, 2024) over $[-\omega_{in}, \omega_{in}]$ and $[-\omega_b, \omega_b]$, respectively. Sampling their entries from Gaussian distri-

butions is also popular (Verstraeten et al., 2007). The entries of $\mathbf{W}_h$ are randomly sampled from a uniform distribution over $[-1, 1]$ and then rescaled to have a desired spectral radius $\rho$[3]. In practical applications, the spectral radius is generally constrained to be smaller than $1$.

The final output is retrieved through the linear readout:

$$\mathbf{y}_t = \mathbf{W}_{out}\mathbf{h}_t + \mathbf{b}_{out}. \quad (2)$$

where $\mathbf{y}_t \in \mathbb{R}^{N_{out}}$ denotes the network output at time step $t$, and $\mathbf{W}_{out} \in \mathbb{R}^{N_{out} \times N_h}$ and $\mathbf{b}_{out} \in \mathbb{R}^{N_{out}}$ denote the readout weight matrix and bias vector, respectively. The readout is typically optimized via lightweight closed-form solutions, e.g. ridge regression or least squares methods.

The model's state update function defined in (1) may depend, in principle, on the initial point $\mathbf{h}_0$. This leads to non-deterministic behaviors, i.e. the impossibility to determine the state solely from the inputs. To avoid that, the reservoir in ESNs is initialized subject to the Echo State Property (ESP) (Yildiz et al., 2012), a useful stability condition for guiding ESN initialization. We say that a discrete time dynamical system defined by the transition function $F(\mathbf{h}, \mathbf{x})$ satisfies the ESP if the states asymptotically depends only on the system's inputs. In other words, for any

---

[3]The spectral radius of a matrix $\mathbf{A}$, denoted $\rho(\mathbf{A})$, is defined as the largest among the lengths of its eigenvalues.

two initial points $\mathbf{h}_0$ and $\mathbf{h}_0'$, then

$$\lim_{t \to \infty} \|F(\mathbf{h}_{t\text{-}1}, \mathbf{x}_t) - F(\mathbf{h}_{t-1}', \mathbf{x}_t)\|_2 = 0. \qquad (3)$$

Several ESN variants lay their foundation at the intersection of the RC and DL frameworks. In particular, Deep Echo State Networks (DeepESNs) (Gallicchio et al., 2017) generalize the concept of shallow ESNs towards deep architectural constructions, where multiple untrained reservoirs are stacked on top of each other. The increased feed-forward depth has been shown to provide advantageous architectural bias relative to shallow ESNs. More recently, Residual Echo State Networks (ResESNs) (Ceni & Gallicchio, 2024) introduced orthogonal residual connections along the temporal dimension to enhance long-term information processing capabilities. Other works have explored structured transforms to improve the efficiency of RC, including Simple Cycle Reservoir (SCR) (Rodan & Tino, 2011), where the transition matrix employs a fixed ring topology, and Structured Reservoir Computing (Structured RC) (Dong et al., 2020), where the transition matrix consists of a composition of Hadamard and diagonal matrices.

**Universality of linear reservoirs.** ESP guarantees the universality of the reservoir system in approximating fading memory filters (Grigoryeva & Ortega, 2018a). Provided that the ESP holds, similar universality results have also been proven for reservoirs characterized by linear dynamics, when combined with non-linear readouts that universally approximate functions $f : \mathbb{C}^{N_h} \to \mathbb{C}^{N_y}$ (Grigoryeva & Ortega, 2018b; Gonon & Ortega, 2020). Therefore, linear recurrence ESNs, in combination with non-linear readouts, can arbitrarily well approximate any time-invariant causal filter with the fading memory property.

**Transformers.** Transformers are the de-facto standard architecture for sequence modeling, managing to replace recurrent models on real-world tasks since their introduction in (Vaswani et al., 2017). Unlike RNNs, transformers are feedforward architectures employing self-attention, enabling access to every position of the sequence at any time. This allows for great parallelization and modeling capacity, but comes at a cost of a quadratic complexity in sequence length, making scaling to long contexts challenging. To address this issue, many variants of self-attention have been proposed to lower the computational and memory burden while preserving expressivity (Tay et al., 2022).

**State Space Models.** Besides training instabilities, another major drawback of stateful sequence models like RNNs is that the input needs to be processed sequentially, greatly limiting the parallelization capabilities of these type of architectures on modern accelerators. One of the most promising approaches for the parallelization of recurrent models is that of (Deep) State Space Models (SSMs) (Gu et al., 2022). SSMs start from the idea of a linear state

space dynamical system, and devise initialization and discretization strategies that help improve memory capacity of the system. Linear recurrence is easily parallelizable (see e.g. Martin & Cundy, 2018), greatly improving training and inference efficiency of this family of models. Structured SSMs such as S4 and S5 (Gu et al., 2022; Smith et al., 2023) demonstrated how linear recurrence, along with a careful initialization based on HiPPO matrices (Gu et al., 2020), can enhance long memory propagation on long sequence tasks. Building on these foundations, Mamba (Gu & Dao, 2024; Dao & Gu, 2024) proposes a selective architecture that allows to change the dynamic based on its inputs. These advances position SSMs as contenders to transformers in sequence modeling, particularly in tasks demanding long memory retention.

**Linear recurrent unit.** Linear recurrent Unit (LRU) (Orvieto et al., 2023) successfully applied linear recurrence to general RNNs, demonstrating that it is possible to deviate from the strict initialization and parametrization rules of SSMs, while retaining their impressive performance. Rather than initializing matrices using HiPPO theory as standard SSMs, LRU employs a diagonal transition matrix, whose eigenvalues are initialized inside the unitary complex disk, using a de-coupled parametrization of their magnitude and phase. In particular, the magnitude is chosen in an interval $[r_{\min}, r_{\max}]$, which allows a finer control over stability and memory capacity of the model. The linear, diagonal structure reduces recurrent updates to parallelizable element-wise operations, and the initialization strategy makes the architecture more stable for longer sequences.

## 3. Parallel Echo State Networks

Inspired by the success of state space modeling, we propose a more scalable and efficient approach for designing reservoirs. We introduce *Parallel Echo State Network (ParalESN)*, a novel class of untrained RNNs based on linear diagonal recurrence in the complex domain, similar to that of LRU. This enables parallel temporal processing via associative scan as well as a reduced memory footprint for the transition matrix. The recurrence is followed by a mixing layer that combines reservoir states, enabling interaction among the components of the hidden state which would otherwise evolve independently due to the diagonal structure of the transition matrix. As is standard in the RC paradigm, the readout layer is the only trainable component. Throughout the paper, we explore a *shallow* and a *deep* variant of ParalESN, which we denote ParalESN and ParalESN (deep), respectively. Fig. 1 illustrates the proposed architecture. Fig. 2 illustrates ParalESN's computational advantage over traditional RC with respect to sequence length (left) and reservoir size (right). ParalESN's recurrence time scales logarithmically with se-

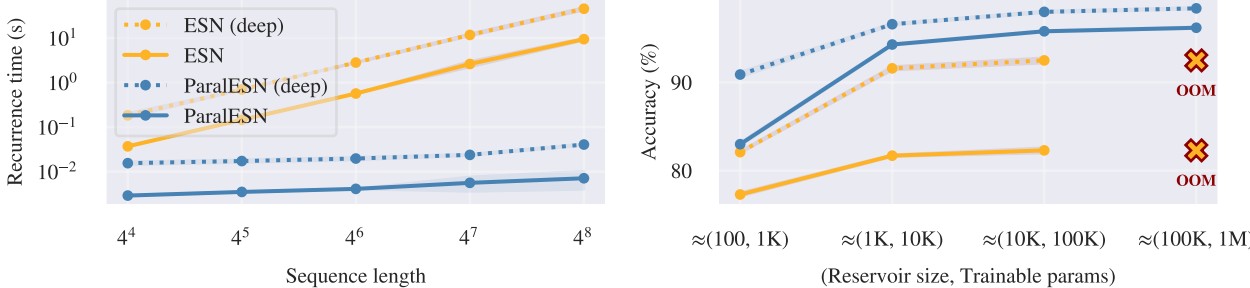

*Figure 2.* (*left*) Time required to perform the recurrence in ParalESN and traditional ESNs for increasing sequence lengths, assuming 128 recurrent neurons and 5 layers for deep configurations. ParalESN scales logarithmically with sequence length, whereas traditional ESNs scale linearly. (*right*) Scaling ParalESN and traditional ESNs to high-dimensional reservoirs on the sMNIST task. Traditional ESNs run out-of-memory (OOM) at approximately 100K reservoir neurons, while ParalESN fits into memory.

quence length, rather than linearly, due to its ability to parallelize the recurrence. Additionally, thanks to memory-efficient parameterizations of its matrices, ParalESN does not scale quadratically with hidden size, enabling higher-dimensional reservoirs [4]. See Appendix B for a thorough computational complexity analysis.

Let $\mathbf{z}_t^{(0)} = \mathbf{x}_t$ denote the input to the first layer, where $\mathbf{x}_t \in \mathbb{R}^{N_{\text{in}}}$ is the external input. The reservoir dynamics at layer $\ell$ are described by:

$$
\begin{aligned}
\mathbf{h}_t^{(\ell)} =& (1 - \tau^{(\ell)}) \, \mathbf{h}_{t\text{-}1}^{(\ell)} \\
&+ \tau^{(\ell)} \left( \mathbf{\Lambda}_{\text{h}}^{(\ell)} \mathbf{h}_{t\text{-}1}^{(\ell)} + \mathbf{W}_{\text{in}}^{(\ell)} \mathbf{z}_t^{(\ell-1)} + \mathbf{b}^{(\ell)} \right),
\end{aligned} \quad (4)
$$

where superscript $(\ell)$ denotes layer-specific hyperparameters and weights. For each time step $t \in \{1, \dots, T\}$, $\mathbf{h}_t^{(\ell)} \in \mathbb{C}^{N_{\text{h}}}$ is the reservoir state and $\mathbf{z}_t^{(\ell)} \in \mathbb{R}^{N_{\text{h}}}$ is the hidden state after the mixing step. The matrix $\mathbf{\Lambda}_{\text{h}}^{(\ell)} \in \mathbb{C}^{N_{\text{h}} \times N_{\text{h}}}$ is a diagonal transition matrix, $\mathbf{b}^{(\ell)} \in \mathbb{C}^{N_{\text{h}}}$ is the bias vector, and $\tau^{(\ell)} \in (0, 1]$ is the leaky rate. Since the recurrence is linear, the leaky integration can be partially absorbed into the transition matrix. Accounting for leakage, the effective transition matrix becomes $\bar{\mathbf{\Lambda}}_{\text{h}}^{(\ell)} = (1 - \tau^{(\ell)})\mathbf{I} + \tau^{(\ell)}\mathbf{\Lambda}^{(\ell)}{}_{\text{h}}$, where $\mathbf{I}$ is the identity matrix. The input weight matrix $\mathbf{W}_{\text{in}}^{(1)} \in \mathbb{C}^{N_{\text{h}} \times N_{\text{in}}}$ is dense for the first layer, mapping the external input to the hidden dimension. For subsequent layers, the input weight matrix $\mathbf{W}_{\text{in}}^{(\ell>1)} \in \mathbb{C}^{N_{\text{h}} \times N_{\text{h}}}$ employs the following ring topology (Rodan & Tino, 2011; Verzelli et al., 2021; Tino, 2020; Pinna et al., 2025) to reduce memory overhead:

$$
\mathbf{W}_{\text{in}}^{(\ell>1)} = \begin{bmatrix} 0 & 0 & \cdots & 0 & w_1 \\ w_2 & 0 & \cdots & 0 & 0 \\ 0 & w_3 & \cdots & 0 & 0 \\ \vdots & \vdots & \ddots & \vdots & \vdots \\ 0 & 0 & \cdots & w_{N_{\text{h}}} & 0 \end{bmatrix}, \quad (5)
$$

---

[4]Traditional RC generally employs dense parameterizations.

The matrix in (5) shifts the input vector and then applies an element-wise scaling. Consequently, we only need to store a $N_{\text{h}}$-dimensional vector of scaling coefficients, significantly reducing the memory footprint in deeper layers. The entries of the input weight matrices are initialized by sampling the real and imaginary parts independently from a uniform distribution over $[-1, 1]$ and then scaling each row $i$ of the matrix by $\sqrt{1 - |\lambda_i|^2}$, where $\lambda_i$ is the corresponding diagonal element (eigenvalue) of the transition matrix $\bar{\mathbf{\Lambda}}_{\text{h}}^{(\ell)}$. The bias vectors are sampled from a uniform distribution and scaled by hyperparameter $\omega_{\text{b}}^{(\ell)}$. The diagonal elements of the transition matrix are initialized to control the eigenvalue distribution, with spectral radii sampled uniformly from $[\rho_{\min}^{(\ell)}, \rho_{\max}^{(\ell)}]$ and phases from $[\theta_{\min}^{(\ell)}, \theta_{\max}^{(\ell)}]$, forming complex eigenvalues $\rho^{(\ell)} e^{i\theta^{(\ell)}}$.

The mixing function $f_{\text{mix}}^{(\ell)}$ is defined as:

$$
\mathbf{z}_t^{(\ell)} = f_{\text{mix}}^{(\ell)}(\mathbf{h}_t^{(\ell)}) = \tanh \left( \Re \left( \mathbf{W}_{\text{mix}}^{(\ell)} * \mathbf{h}_t^{(\ell)} + \mathbf{b}_{\text{mix}}^{(\ell)} \right) \right), \quad (6)
$$

where $*$ denotes the convolution operator, $\mathbf{W}_{\text{mix}}^{(\ell)} \in \mathbb{C}^k$ is a 1-D kernel of size $k^{(\ell)}$, $\mathbf{b}_{\text{mix}}^{(\ell)} \in \mathbb{C}$ is a scalar bias, and $\Re(\cdot)$ extracts the real part. The kernel slides across the hidden dimension of $\mathbf{h}_t^{(\ell)} \in \mathbb{C}^{N_{\text{h}}}$ with same-padding, producing an output of identical dimension. We employ a 1-D convolution rather than a dense matrix to reduce the memory footprint of the mixing function. The same kernel is shared across all time steps, requiring only $k + 1$ parameters regardless of sequence length or hidden size. The kernel and bias entries are sampled randomly from a uniform distribution over $[-\omega_{\text{mix}}^{(\ell)}, \omega_{\text{mix}}^{(\ell)}]$ and $[-\omega_{\text{mixb}}^{(\ell)}, \omega_{\text{mixb}}^{(\ell)}]$, respectively.

The readout aggregates the mixed states from all $L$ layers:

$$
\mathbf{y}_t = f_{\text{readout}}(\mathbf{z}_t^{(1)}, \dots, \mathbf{z}_t^{(L)}). \quad (7)
$$

For classification tasks, only the final time step is used, $\mathbf{y} = f_{\text{readout}}(\mathbf{z}_t^{(1)}, \dots, \mathbf{z}_t^{(L)})$.

## 4. Theoretical Analysis

Here, we derive a simple condition for the ESP to hold, which is necessary to prove a universality result for the class of linear reservoir models (Grigoryeva & Ortega, 2018a). For simplicity, we consider a one-layer ParalESN. The sequence of hidden states produced by the model, given a sequence of inputs $\{\mathbf{x}_1, \ldots, \mathbf{x}_t\} \in (\mathbb{C}^{N_{in}})^T$ and a starting point $\mathbf{h}_0 \in \mathbb{C}^{N_h}$, is $(\mathbf{h}_0, ..., \mathbf{h}_T)$, with

$$\mathbf{h}_t = \begin{cases} \mathbf{h}_0 & \text{if } t = 0 \\ \bar{\mathbf{\Lambda}}_h \mathbf{h}_{t\text{-}1} + \tau(\mathbf{W}_{in}\mathbf{x}_t + \mathbf{b}) & \text{otherwise} \end{cases} \quad (8)$$

$$\mathbf{y}_t = f_{out}(\mathbf{h}_t), \quad (9)$$

where $f_{out}$ is the readout function. Because the recurrence is linear, the readout function must be non-linear to preserve expressivity. This definition differs slightly from equations (4), (6), and (7), but since training does not come into play in this section, we can incorporate $f_{mix}$ and $f_{readout}$ into a single function $f_{out}$.

### 4.1. Echo State Property

We derive a simple condition for the ESP to hold in the case of diagonal linear ESNs as in (8). The same condition on the spectral radius that is necessary for the ESP of ESNs (see, e.g., (Jaeger & Haas, 2004)) is also sufficient in our case. Moreover, in the case of a diagonal transition matrix, we can directly control the spectral radius by the largest diagonal element in absolute value.

**Theorem 4.1** (Sufficient and necessary conditions for the ESP). *A ParalESN has the ESP if and only if the diagonal elements $\lambda_1, ..., \lambda_{N_h}$ of the transition matrix $\bar{\mathbf{\Lambda}}_h$ are such that for each $i$, $|\lambda_i| < 1$, where $|\cdot|$ is the complex modulus.*

The proof is given in Appendix D.1.

### 4.2. Expressivity of ParalESN

An ESN with linear recurrence, MLP readout, and the ESP is universal in the family of fading memory filters, and in particular, it is as expressive as non-linear ESNs that satisfy the same property (Grigoryeva & Ortega, 2018a;b). See Appendix C for definitions on fading memory filters. Having established the conditions for ParalESN to have the ESP in Theorem 4.1, we now show that ParalESN is as expressive as an ESN with linear recurrence and any arbitrary transition matrix $\mathbf{W}_h \in \mathbb{C}^{N_h \times N_h}$ and MLP readout.

**Proposition 4.2.** *Let $\mathbf{W}_h$ be a matrix with independent and identically distributed entries $w_{i,j} \sim p(w)$, where $p(w)$ is a density [...] Consider an ESN with linear recurrence and a 1-layer MLP readout, defined by*

$$\begin{cases} \mathbf{h}_t = \mathbf{W}_h \mathbf{h}_{t\text{-}1} + \mathbf{W}_{in}\mathbf{x}_t \\ \mathbf{y}_t = \mathbf{W}_{out}(\sigma(\mathbf{W}_h \mathbf{h}_t)) = MLP(\mathbf{h}_t) \end{cases} \quad (10)$$

*Then, with probability 1, there exists a ParalESN with MLP readout, defined by* (8) *and* (9) *with $f_{out}$ being an MLP, such that for any given input, the two models produce the same output.*

The proposition can be proven by diagonalizing $\mathbf{W}_h$. The proof is given in Appendix D.2.

By Proposition 4.2 and prior results on equivalent expressiveness between ESNs with linear recurrence and standard ESNs, we can deduce that ParalESN and standard ESNs (both satisfying the ESP) are *equivalently expressive*. We state this formally in the following theorem.

**Theorem 4.3.** *The class of ParalESN models with the ESP, endowed with an MLP readout, is universal in the family of fading memory filters.*

In practice, in our experiments, the MLP readout is often decomposed into two layers: the first layer, together with the non-linearity, is treated as the fixed mixing function, and the last layer is treated as the trainable readout.

## 5. Experiments

We empirically evaluate the proposed approach on time series regression and classification tasks (see also Appendix E) against traditional RC and fully-trainable sequence models (see also Appendix F.2). See Appendix F for details on the experimental setting and model selection, and Appendix G for additional experiments, including hyperparameter sensitivity analyses, ablations, and benchmarks on Long Range Arena (Tay et al., 2021).

**Comparison with respect to traditional RC.** Fig. 3 compares ParalESN with respect to traditional RC from predictive performance and computational efficiency perspectives. The left panel presents the trade-off between predictive performance and computational efficiency; the right panel presents a critical difference diagram ranking models based on their overall performance. We observe that ParalESN and ParalESN (deep) are both more accurate and more efficient than their counterparts. In particular, ParalESN (deep), despite consisting of multiple reservoir layers, remains competitive in terms of recurrence speed with a single layer ESN. Finally, ParalESN outperforms its shallow counterpart by a statistically significant margin. Although there is no statistically significant difference between ParalESN (deep) and ESN (deep) in terms of performance, the former is the top-performing model while being considerably more efficient.

### 5.1. Time Series Regression

Memory-based tasks are designed to assess the ability to recall delayed versions of the input. We consider Mem-Cap (Jaeger, 2002), ctXOR (Verstraeten et al., 2010), and

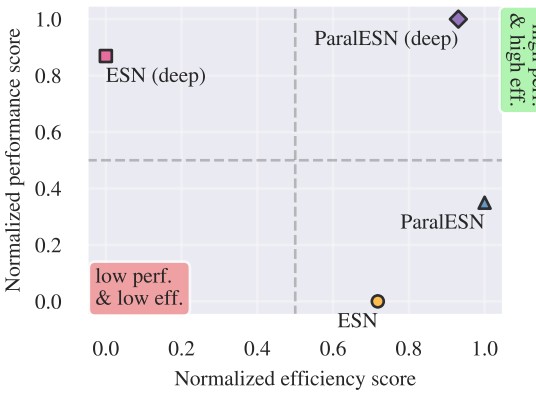
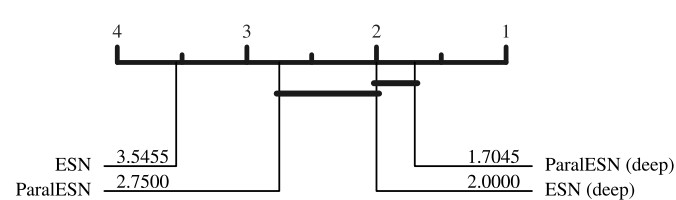

*Figure 3.* (*left*) Analysis of the trade-off between predictive performance (error for regression tasks and accuracy for classification tasks) and computational efficiency (training time) for ParalESN and traditional ESNs across all considered benchmarks. For each model, we compute the percentage improvement over the ESN baseline for each task. The normalized scores are then obtained via min-max normalization of the average improvements, mapping them to a $[0, 1]$ scale, where 0 corresponds to the worst-performing model and 1 to the best-performing one. Overall, ParalESN and ParalESN (deep) outperform their counterparts while being more efficient. (*right*) Critical difference diagram computed via a Wilcoxon test (Demšar, 2006), showing the average rank (lower is better). Models are ranked based on their overall performance across all benchmarks. Cliques (solid lines) connect models for which there is no statistically significant difference in performance. On average, ParalESN (deep) is the top-performing model.

SinMem (Inubushi & Yoshimura, 2017). Forecasting tasks evaluate the model's ability to predict future time steps. We consider Lorenz96 (Lorenz, 1996), Mackey-Glass (MG) (Jaeger & Haas, 2004), NARMA, and a selection of real-world time series from (Zhou et al., 2021), including ETTh1/2 and ETTm1/2.

**Discussion.** Table 1 and Table 2 present the test set results on memory-based and forecasting tasks, respectively. Fig. 4 compares the training time of ParalESN to that of traditional ESNs. Our experiments demonstrate that ParalESN achieves results comparable to traditional RC across a wide range of time series regression benchmarks, while offering substantial advantages in computational efficiency[5] Across all benchmarks, ParalESN trains an entire order of magnitude faster, except for Lorenz25 and Lorenz50, where the relatively small sequence length reduces the benefit of parallelizing the recurrence. Observe that even ParalESN (deep), despite consisting of multiple reservoir layers, trains faster than a traditional, shallow ESN consisting of a single layer. Indeed, while the readout layer is trained via Ridge regression in both cases, the time required to process the sequence through the untrained reservoir is significantly lower in the proposed approach compared to traditional ESNs, thanks to parallel sequence processing.

---

[5]In PyTorch (used in our experiments), complex-valued arithmetic is generally not yet as optimized as its real-valued counterpart. The computational efficiency advantage of ParalESN is therefore measured under conservative conditions: with more optimized complex arithmetic libraries, this advantage would likely be even greater.

### 5.2. Time Series Classification

The time series classification benchmark consists of a selection of tasks from the UEA & UCR repository (Bagnall et al., 2018; Dau et al., 2019). For 1-D pixel-level classification, we consider two variants of the MNIST dataset (LeCun, 1998): (i) sequential MNIST (sMNIST), where pixels are flattened into a one-dimensional sequence, and (ii) permuted sequential MNIST (psMNIST), where a random permutation is additionally applied to the flattened pixels.

**Discussion.** Tables 3 and 4 present test set results on time series classification tasks from the UEA & UCR repository and on sMNIST and psMNIST, respectively. Fig. 5 visualizes the trade-off between predictive performance and computational efficiency across all models for the MNIST benchmarks. On time series classification, ParalESN achieves higher test accuracy compared to a shallow ESN: +27.9% on Blink, +3.7% on FaultDetectionA, +7.6% on FordA, +5.7% on FordB, and +3.3% on StarLightCurves. Similarly, ParalESN (deep) outperforms ESN (deep) by +10.0% on Blink, +8.8% on FaultDetectionA, +2.7% on FordA, +0.7% on FordB, and +2.3% on StarLightCurves. On sMNIST and psMNIST, ParalESN achieves higher test accuracy by +13.4% and +17.1%, respectively, compared to traditional ESN. ParalESN (deep) provides gains of +7.7% and +13.1% over ESN (deep). These performance improvements come alongside substantial efficiency gains, with ParalESN and ParalESN (deep) requiring half or less of the training time, $CO_2$ emissions, and energy of traditional RC. Additionally, unlike traditional RC, ParalESN is competitive with LSTM, Transformer, S4, LRU, and Mamba.

*Table 1.* Test set results on memory-based tasks, assuming 128 recurrent neurons for each model. The **best result** is highlighted in bold.

| MEMORY-BASED | ↑ MEMCAP | $\cdot 10^{-1}$ ↓ CTXOR5 | $\cdot 10^{-1}$ ↓ CTXOR10 | $\cdot 10^{-1}$ ↓ SINMEM10 | $\cdot 10^{-1}$ ↓ SINMEM20 |
|---|---|---|---|---|---|
| ESN | $50.6_{\pm 1.6}$ | $3.6_{\pm 0.1}$ | $7.7_{\pm 0.6}$ | $3.6_{\pm 0.1}$ | $3.7_{\pm 0.1}$ |
| ESN (deep) | $56.8_{\pm 1.3}$ | $\mathbf{3.4_{\pm 0.2}}$ | $\mathbf{5.2_{\pm 1.0}}$ | $1.2_{\pm 0.1}$ | $\mathbf{1.6_{\pm 0.1}}$ |
| ParalESN | $114.5_{\pm 1.4}$ | $3.9_{\pm 0.1}$ | $8.2_{\pm 0.2}$ | $3.7_{\pm 0.0}$ | $3.7_{\pm 0.0}$ |
| ParalESN (deep) | $\mathbf{125.0_{\pm 0.2}}$ | $3.6_{\pm 0.1}$ | $5.6_{\pm 0.4}$ | $\mathbf{1.0_{\pm 0.2}}$ | $2.5_{\pm 0.4}$ |

*Table 2.* Test set results on forecasting tasks, assuming 128 recurrent neurons for each model. The **best result** is highlighted in bold.

| FORECASTING | $\cdot 10^{-2}$ ↓ Lz25 | $\cdot 10^{-2}$ ↓ Lz50 | $\cdot 10^{-4}$ ↓ MG | $\cdot 10^{-2}$ ↓ MG84 | $\cdot 10^{-2}$ ↓ N10 | $\cdot 10^{-2}$ ↓ N30 | $\cdot 10^{-1}$ ↓ ETTH1 | $\cdot 10^{-1}$ ↓ ETTH2 | $\cdot 10^{-1}$ ↓ ETTM1 | $\cdot 10^{-1}$ ↓ ETTM2 |
|---|---|---|---|---|---|---|---|---|---|---|
| ESN | $10.0_{\pm 0.3}$ | $30.8_{\pm 0.6}$ | $3.0_{\pm 0.0}$ | $6.5_{\pm 0.4}$ | $\mathbf{2.7_{\pm 0.4}}$ | $10.3_{\pm 0.1}$ | $9.1_{\pm 0.2}$ | $13.0_{\pm 1.7}$ | $6.7_{\pm 0.1}$ | $9.9_{\pm 7.3}$ |
| ESN (deep) | $\mathbf{9.7_{\pm 0.2}}$ | $30.5_{\pm 0.3}$ | $2.0_{\pm 0.0}$ | $4.2_{\pm 0.2}$ | $3.0_{\pm 0.5}$ | $10.1_{\pm 0.1}$ | $8.9_{\pm 0.1}$ | $\mathbf{9.6_{\pm 0.5}}$ | $6.6_{\pm 0.0}$ | $6.0_{\pm 0.6}$ |
| ParalESN | $10.4_{\pm 0.5}$ | $30.2_{\pm 0.5}$ | $2.8_{\pm 0.2}$ | $7.4_{\pm 0.4}$ | $3.7_{\pm 0.8}$ | $10.2_{\pm 0.1}$ | $9.0_{\pm 0.2}$ | $12.8_{\pm 1.2}$ | $\mathbf{6.5_{\pm 0.1}}$ | $5.1_{\pm 0.1}$ |
| ParalESN (deep) | $10.3_{\pm 0.3}$ | $\mathbf{29.4_{\pm 0.3}}$ | $2.6_{\pm 0.4}$ | $5.2_{\pm 0.5}$ | $4.5_{\pm 1.0}$ | $10.2_{\pm 0.2}$ | $\mathbf{8.8_{\pm 0.1}}$ | $9.7_{\pm 0.9}$ | $\mathbf{6.5_{\pm 0.0}}$ | $\mathbf{5.0_{\pm 0.0}}$ |

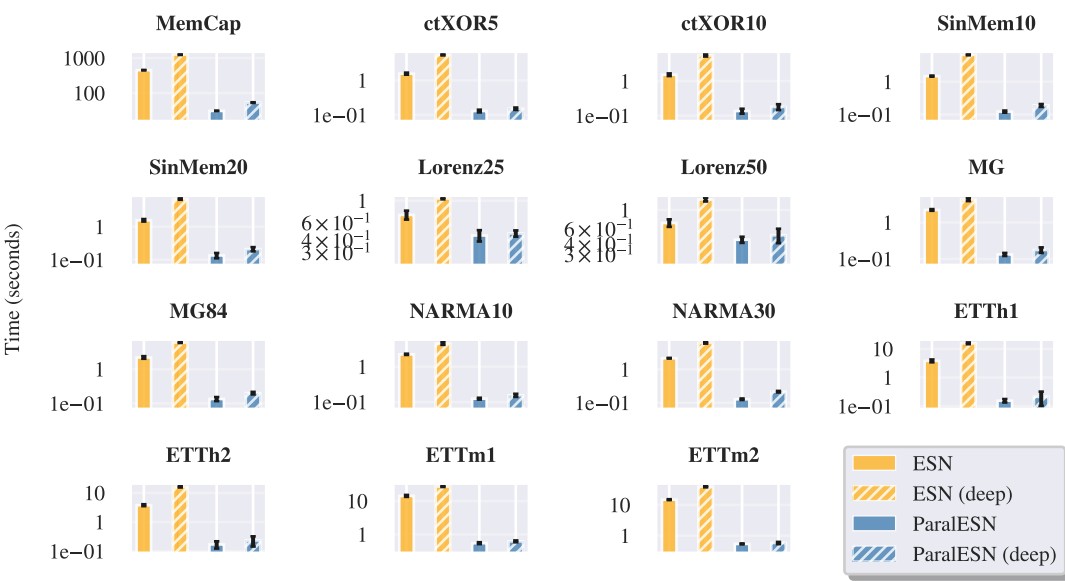

*Figure 4.* Training time comparison between ParalESNs and traditional ESNs for each memory-based and forecasting benchmark. ParalESN and ParalESN (deep) train orders of magnitudes faster.

# 6. Conclusions

We introduced ParalESN, a novel framework for constructing high-dimensional, efficient, and parallelizable untrained RNNs based on linear diagonal recurrence. This work addresses fundamental limitations of traditional RC, including the need to process temporal data sequentially and the prohibitive memory footprint of high-dimensional reservoirs. Results across various time series and 1-D pixel-level classification benchmarks demonstrate that ParalESN is, on average, more accurate and faster than traditional RC, while remaining competitive with fully-trainable sequence models at a fraction of their computational cost.

# Acknowledgements

This work has been supported by EMERGE, a project funded by the European Innovation Council (prj. code 101070918), and by NEURONE, a project funded by the European Union - Next Generation EU, M4C1 CUP I53D23003600006, under program PRIN 2022 (prj. code 20229JRTZA). Computational resources were provided by Computing@Unipi, a computing service of the University of Pisa.

*Table 3.* Test set results on time series classification tasks, assuming 1024 recurrent neurons for each model. The **best result** is highlighted in bold.

| CLASSIFICATION | ↑ BLINK | ↑ FAULTDETECTIONA | ↑ FORDA | ↑ FORDB | ↑ STARLIGHTCURVES |
|---|---|---|---|---|---|
| ESN | $68.9_{\pm 5.5}$ | $71.3_{\pm 0.8}$ | $75.8_{\pm 0.9}$ | $62.7_{\pm 0.8}$ | $92.1_{\pm 0.7}$ |
| ESN (DEEP) | $86.5_{\pm 1.8}$ | $86.0_{\pm 0.6}$ | $90.1_{\pm 0.8}$ | $76.1_{\pm 0.8}$ | $93.8_{\pm 0.4}$ |
| PARALESN | $\mathbf{96.8_{\pm 1.1}}$ | $75.0_{\pm 0.4}$ | $83.4_{\pm 0.7}$ | $68.4_{\pm 1.0}$ | $95.4_{\pm 0.4}$ |
| PARALESN (DEEP) | $96.5_{\pm 0.4}$ | $\mathbf{94.8_{\pm 0.9}}$ | $\mathbf{92.8_{\pm 0.5}}$ | $\mathbf{76.8_{\pm 1.6}}$ | $\mathbf{96.1_{\pm 0.3}}$ |

*Table 4.* Test set results on sMNIST and psMNIST tasks. The **best result** is highlighted in bold, second-best is underlined.

|  |  |  | **sMNIST** |  |  |
|---|---|---|---|---|---|
| MODEL | PARAMS. | ↑ ACCURACY | ↓ TIME (MIN.) | ↓ EMISSIONS (KG) | ↓ ENERGY (KWH) |
| LSTM | $\approx 160k$ | $97.5_{\pm 1.4}$ | $80.8_{\pm 6.8}$ | $0.34_{\pm 0.14}$ | $1.02_{\pm 0.42}$ |
| TRANSFORMER | $\approx 160k$ | $98.4_{\pm 0.1}$ | $141.0_{\pm 14.1}$ | $0.60_{\pm 0.28}$ | $1.81_{\pm 0.86}$ |
| S4* | $\approx 160k$ | $\mathbf{99.2_{\pm 0.0}}$ | $16.1_{\pm 0.0}$ | $0.53_{\pm 0.0}$ | $1.61_{\pm 0.0}$ |
| LRU | $\approx 160k$ | $\underline{98.5_{\pm 0.2}}$ | $29.1_{\pm 1.85}$ | $0.18_{\pm 0.02}$ | $0.57_{\pm 0.05}$ |
| MAMBA | $\approx 200k$ | $98.4_{\pm 0.1}$ | $22.87_{\pm 4.48}$ | $0.19_{\pm 0.02}$ | $0.57_{\pm 0.06}$ |
| ESN | $\approx 160k$ | $82.5_{\pm 7}$ | $4.3_{\pm 0.1}$ | $\underline{0.02_{\pm 0.00}}$ | $0.07_{\pm 0.00}$ |
| ESN (DEEP) | $\approx 160k$ | $91.4_{\pm 1.1}$ | $8.8_{\pm 0.1}$ | $0.04_{\pm 0.00}$ | $0.13_{\pm 0.00}$ |
| PARALESN | $\approx 160k$ | $96.2_{\pm 1.3}$ | $\mathbf{2.7_{\pm 0.7}}$ | $\mathbf{0.01_{\pm 0.00}}$ | $\mathbf{0.04_{\pm 0.10}}$ |
| PARALESN (DEEP) | $\approx 160k$ | $97.2_{\pm 0.2}$ | $\underline{3.3_{\pm 0.5}}$ | $\underline{0.02_{\pm 0.00}}$ | $\underline{0.05_{\pm 0.00}}$ |

|  |  |  | **psMNIST** |  |  |
|---|---|---|---|---|---|
| MODEL | PARAMS. | ↑ ACCURACY | ↓ TIME (MIN.) | ↓ EMISSIONS (KG) | ↓ ENERGY (KWH) |
| LSTM | $\approx 160k$ | $92.8_{\pm 0.5}$ | $89.3_{\pm 4.2}$ | $0.47_{\pm 0.04}$ | $1.41_{\pm 0.11}$ |
| TRANSFORMER | $\approx 160k$ | $97.4_{\pm 0.2}$ | $156.8_{\pm 2.7}$ | $0.65_{\pm 0.24}$ | $1.98_{\pm 0.73}$ |
| S4* | $\approx 160k$ | $\mathbf{97.9_{\pm 0.0}}$ | $9.87_{\pm 0.0}$ | $0.35_{\pm 0.0}$ | $1.07_{\pm 0.0}$ |
| LRU | $\approx 160k$ | $\underline{97.8_{\pm 0.1}}$ | $33.8_{\pm 3.42}$ | $0.21_{\pm 0.03}$ | $0.63_{\pm 0.09}$ |
| MAMBA | $\approx 200k$ | $92.6_{\pm 0.1}$ | $43.25_{\pm 5.02}$ | $0.24_{\pm 0.03}$ | $0.73_{\pm 0.08}$ |
| ESN | $\approx 160k$ | $78.2_{\pm 1.6}$ | $4.3_{\pm 0.1}$ | $\underline{0.02_{\pm 0.00}}$ | $0.06_{\pm 0.00}$ |
| ESN (DEEP) | $\approx 160k$ | $82.1_{\pm 3.7}$ | $7.3_{\pm 0.1}$ | $0.04_{\pm 0.00}$ | $0.11_{\pm 0.00}$ |
| PARALESN | $\approx 160k$ | $96.9_{\pm 0.1}$ | $\mathbf{2.8_{\pm 0.3}}$ | $\mathbf{0.01_{\pm 0.00}}$ | $\mathbf{0.04_{\pm 0.00}}$ |
| PARALESN (DEEP) | $\approx 160k$ | $95.2_{\pm 0.1}$ | $\underline{3.1_{\pm 0.2}}$ | $\underline{0.02_{\pm 0.00}}$ | $\underline{0.05_{\pm 0.00}}$ |

\* The computational efficiency metrics reported for S4 are likely underestimates, as a different and more powerful GPU was used for this model due to hardware availability constraints.

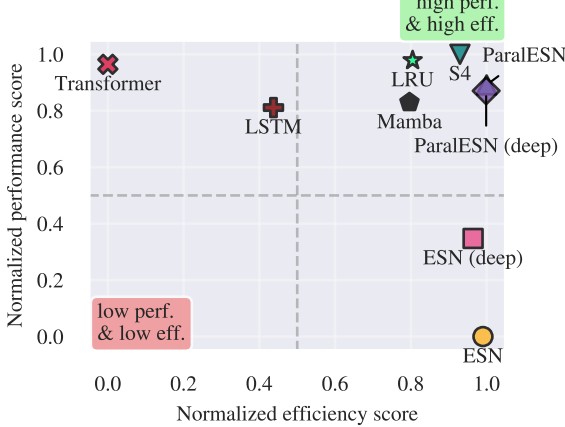

*Figure 5.* Trade-off between performance (test accuracy) and efficiency (training time) for ParalESN, traditional RC, and fully-trainable sequence models on the sMNIST and psMNIST benchmarks. For each model and benchmark, we compute the percentage improvement over the ESN baseline. The normalized scores are then obtained via min-max normalization of the average improvements, mapping them to a $[0, 1]$ scale where 0 corresponds to the worst-performing model and 1 to the best-performing one. ParalESN is competitive with fully-trainable models while delivering substantial efficiency improvements.

## Impact Statement

This paper presents work whose goal is to advance the field of Machine Learning. There are many potential societal consequences of our work, none which we feel must be specifically highlighted here.

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

## A. Table of contents for the Appendix

The Appendix is organized as follows:

## B. Computational Complexity

Here, we compare the time and space complexity of ParalESN, traditional ESNs, and other RC approaches based on structured transforms, including SCR (Rodan & Tino, 2011) and Structured RC (Dong et al., 2020). For simplicity, we consider shallow, single-layer architectures and omit the computational cost of training the readout, as this cost is the same for all models. Table 5 provides an overview of the computational complexity analysis. In terms of time complexity, the diagonal transition matrix of ParalESN yields a number of operations that scales linearly in the reservoir size $N_\mathrm{h}$. Furthermore, the linear recurrence enables all time steps to be computed in parallel via an associative scan, removing the linear dependency on the sequence length and reducing it to logarithmic. In terms of space complexity, ParalESN significantly reduces the memory footprint of its parameters compared to traditional ESNs by employing a diagonal transition matrix, storing only $N_\mathrm{h}$ diagonal elements rather than a full $N_\mathrm{h} \times N_\mathrm{h}$ matrix. This is crucial for enabling the construction of larger reservoirs.

**ESN.** We denote by $\{x_1, \ldots, x_\mathrm{T}\}$ an input sequence of length $T$, where each $x_\mathrm{i} \in \mathbb{R}^{N_\mathrm{in}}$. A standard nonlinear ESN with $N_\mathrm{h}$ hidden units updates its reservoir state according to Equation 1. Each time step requires two matrix-vector multiplications: one with the transition matrix ($N_\mathrm{h} \times N_\mathrm{h}$) and one with the input weight matrix ($N_\mathrm{h} \times N_\mathrm{in}$). Thus, the time complexity for processing the entire sequence is $\mathcal{O}\big(T(N_\mathrm{h}^2 + N_\mathrm{h}N_\mathrm{in})\big) = \mathcal{O}\big(TN_\mathrm{h}^2\big)$, where the equality assumes $N_\mathrm{in} < N_\mathrm{h}$, as is standard in ESNs. Regarding space complexity, ESNs store a full $N_\mathrm{h} \times N_\mathrm{h}$ transition matrix, a full $N_\mathrm{h} \times N_\mathrm{in}$ input weight matrix, and an $N_\mathrm{h}$-dimensional bias vector, giving a parameter space complexity of $\mathcal{O}(N_\mathrm{h}(N_\mathrm{h} + N_\mathrm{in}))$. The space complexity for the recurrence is $\mathcal{O}(TN_\mathrm{h})$, as $T$ hidden states must be stored[6].

**SCR.** The transition matrix employs a ring topology. The state update can be implemented in $\mathcal{O}(N_\mathrm{h})$ time, including the multiplication of the state by a scaling factor $\rho$, giving a total time complexity of $\mathcal{O}(TN_\mathrm{h}N_\mathrm{in})$, the same as a sequential ParalESN. Regarding space complexity, SCR stores only a full $N_\mathrm{h} \times N_\mathrm{in}$ input weight matrix and an $N_\mathrm{h}$-dimensional bias vector. The transformation applied by the ring-topology transition matrix amounts to a cyclic shift of the vector elements and does not need to be stored explicitly. The parameter space complexity is thus $\mathcal{O}(N_\mathrm{h}N_\mathrm{in})$, and the space complexity for the recurrence is the same as in traditional ESNs, $\mathcal{O}(TN_\mathrm{h})$.

**Structured RC.** The full transition matrix $\mathbf{W}_\mathrm{h}$ in (1) is replaced by a composition of Hadamard and diagonal matrices, reducing the time complexity by a logarithmic factor in the reservoir size and achieving $\mathcal{O}(TN_\mathrm{h} \log N_\mathrm{h})$, assuming $N_\mathrm{in} < \log N_\mathrm{h}$. If the Hadamard matrix is stored explicitly, the parameter space complexity is the same as in traditional ESNs, $\mathcal{O}(N_\mathrm{h}(N_\mathrm{h} + N_\mathrm{in}))$, though each element requires only a single bit rather than full floating-point precision since the matrix is binary. Alternatively, if the Hadamard transform is applied algorithmically via the fast Hadamard transform, no explicit storage is required, and only the diagonal matrices and input weights need to be stored, yielding a parameter space complexity of $\mathcal{O}(N_\mathrm{h}N_\mathrm{in})$. The space complexity for the recurrence is the same as in traditional ESNs, $\mathcal{O}(TN_\mathrm{h})$.

**ParalESN.** In our approach, the reservoir update uses a diagonal transition matrix, which reduces the per-step computation to $\mathcal{O}(N_\mathrm{h}N_\mathrm{in})$. Hence, the sequential computation over a length-$T$ sequence has total cost $\mathcal{O}(TN_\mathrm{h}N_\mathrm{in})$. Furthermore, because the state updates are linear, the computation can be parallelized along the temporal dimension. Using a parallel associative scan algorithm (Martin & Cundy, 2018), the dependence on $T$ is reduced to a logarithmic factor[7], yielding an

---

[6]Assuming the case where all time steps are of interest.
[7]Assuming $\Theta(T/\log T)$ parallel processors.

*Table 5.* Overview of time and space complexity.

| | TIME COMPLEXITY | | SPACE COMPLEXITY | |
| MODEL | SINGLE STEP | WHOLE SEQUENCE | PARAMETERS | RECURRENCE |
| --- | --- | --- | --- | --- |
| ESN | $\mathcal{O}(N_{\mathrm{h}}^2)$ | $\mathcal{O}(TN_{\mathrm{h}}^2)$ | $\mathcal{O}(N_{\mathrm{h}}(N_{\mathrm{h}} + N_{\mathrm{in}}))$ | $\mathcal{O}(TN_{\mathrm{h}})$ |
| SCR | $\mathcal{O}(N_{\mathrm{h}}N_{\mathrm{in}})$ | $\mathcal{O}(TN_{\mathrm{h}}N_{\mathrm{in}})$ | $\mathcal{O}(N_{\mathrm{h}}N_{\mathrm{in}})$ | $\mathcal{O}(TN_{\mathrm{h}})$ |
| STRUCTURED RC | $\mathcal{O}(N_{\mathrm{h}} \log N_{\mathrm{h}})$ | $\mathcal{O}(TN_{\mathrm{h}} \log N_{\mathrm{h}})$ | $\mathcal{O}(N_{\mathrm{h}}N_{\mathrm{in}})$ | $\mathcal{O}(TN_{\mathrm{h}})$ |
| PARALESN | $\mathcal{O}(N_{\mathrm{h}}N_{\mathrm{in}})$ | $\mathcal{O}(\log(T)N_{\mathrm{h}}N_{\mathrm{in}})$ | $\mathcal{O}(N_{\mathrm{h}}N_{\mathrm{in}})$ | $\mathcal{O}(TN_{\mathrm{h}})$ |

overall complexity of $\mathcal{O}\left(\log(T)\,N_{\mathrm{h}}N_{\mathrm{in}}\right)$. ParalESN stores the $N_{\mathrm{h}}$ elements of its diagonal transition matrix, an $N_{\mathrm{h}} \times N_{\mathrm{in}}$ input weight matrix, and an $N_{\mathrm{h}}$-dimensional bias vector, giving a parameter space complexity of $\mathcal{O}(N_{\mathrm{h}}N_{\mathrm{in}})$. The space complexity for the recurrence is the same as in traditional ESNs, $\mathcal{O}(TN_{\mathrm{h}})$.

Note that the associative scan could in principle be applied to any linear recurrence RC model, but practical limitations remain. Since the algorithm involves repeatedly squaring the transition matrix, it would generally require $N_{\mathrm{h}}^3$ operations per scan step for full, unstructured matrices. Moreover, in the case of Structured RC, matrix-matrix multiplication would produce a full matrix, destroying the Hadamard structure.

## C. Filters and Fading Memory Property

Here, we provide a brief overview of the definitions used to characterize the class of filters that ESNs and linear ESNs can universally approximate. Specifically, we introduce the concept of fading memory filters. We mainly follow (Grigoryeva & Ortega, 2018a) and refer the reader to it for further details.

**Filters.** Informally, a filter is a function that maps semi-infinite sequences to semi-infinite sequences. More precisely, given $N_{\mathrm{x}} \in \mathbb{N}$ and $N_{\mathrm{y}} \in \mathbb{N}$, we define a filter $\mathcal{U}$ as follows:

$$\mathcal{U} : (\mathbb{R}^{N_{\mathrm{x}}})^{\mathbb{Z}_-} \to (\mathbb{R}^{N_{\mathrm{y}}})^{\mathbb{Z}_-}, \tag{11}$$

where $(\mathbb{R}^N)^{\mathbb{Z}_-}$ is the set of semi-infinite sequences of $N$-dimensional vectors indexed by non-positive integers[8], $\overleftarrow{\mathbf{v}} = (\mathbf{v}_{\mathrm{i}})_{\mathrm{i} \in \mathbb{Z}_-}$ with $\mathbf{v}_{\mathrm{i}} \in \mathbb{R}^N$.

Two desirable properties for filters are *causality* and *time-invariance*. A filter $\mathcal{U}$ is causal if its output at time $t$, $\mathbf{y}_{\mathrm{t}} = \mathcal{U}(\overleftarrow{\mathbf{x}})_{\mathrm{t}}$, depends only on the inputs up to time $t$, i.e., $\overleftarrow{\mathbf{x}}_{(-\infty,t]}$. Causality is an important property for tasks such as forecasting or autoregressive generation: without it, past inputs alone would not be sufficient to uniquely determine the present output. A filter is time-invariant if, given two shifted input sequences, it outputs two sequences shifted by the same amount. Formally, for any positive integer $\tau > 0$, we define the *time-shift operator* $\mathcal{T}_\tau$ by $\mathcal{T}_\tau(\overleftarrow{\mathbf{x}})_{\mathrm{t}} = \overleftarrow{\mathbf{x}}_{\mathrm{t}-\tau}$. A filter is time-invariant if it commutes with this operator. This property ensures that the filter does not depend explicitly on time $t$.

**Infinite norm.** The space $(\mathbb{R}^N)^{\mathbb{Z}}$ can be endowed with a norm, providing a notion of distance. Given the standard Euclidean norm for vectors $\|\cdot\|$, we define the *infinite norm* of a sequence as the supremum of the norms of its elements:

$$\|\overleftarrow{\mathbf{v}}\|_\infty = \sup_{\mathrm{i} \in \mathbb{Z}} \|\mathbf{v}_{\mathrm{i}}\|. \tag{12}$$

A *distance* between two sequences $\overleftarrow{\mathbf{v}}$ and $\overleftarrow{\mathbf{w}}$ is then defined as the infinite norm of their difference:

$$d_\infty(\overleftarrow{\mathbf{v}}, \overleftarrow{\mathbf{w}}) = \|\overleftarrow{\mathbf{v}} - \overleftarrow{\mathbf{w}}\|_\infty. \tag{13}$$

**Weighted norm.** When measuring the difference between two sequences, it is desirable to weight recent values more heavily than distant past ones. The infinite norm treats all time steps equally and therefore cannot capture this requirement. Thus, we introduce a *weighting sequence* $w = (w_i)_{\mathrm{i} \in \mathbb{Z}_-} \in (0,1]^{\mathbb{Z}_-}$ with $\lim_{\mathrm{i} \to -\infty} w_{\mathrm{i}} = 0$, and define the *weighted norm* as:

$$\|\overleftarrow{\mathbf{v}}\|_w = \|w \odot \overleftarrow{\mathbf{v}}\|_\infty, \tag{14}$$

---

[8]Filters can also be defined for bi-infinite sequences, i.e., with inputs and outputs indexed by $\mathbb{Z}$. However, since in practice we are mainly interested in the value of the sequence at the last time step, it is common to restrict the definition to semi-infinite sequences.

where $\odot$ denotes the element-wise product.

**Fading memory property.** We now give a formal definition of the fading memory property. The class of fading memory filters is the family of functions that ESNs, and their linear counterparts, approximate arbitrarily well, thus making them universal approximators. Informally, a fading memory filter is a filter that is continuous with respect to the topology induced by any weighted norm $\| \cdot \|_w$:

**Definition C.1.** A filter $\mathcal{U} : (\mathbb{R}^{N_x})^{\mathbb{Z}-} \to (\mathbb{R}^{N_y})^{\mathbb{Z}-}$ has the *fading memory property* if for any $\overleftarrow{\mathbf{x}}_1 \in (\mathbb{R}^{N_x})^{\mathbb{Z}-}$ and any $\epsilon > 0$, there exists $\delta = \delta(\epsilon) > 0$ such that for any $\overleftarrow{\mathbf{x}}_2 \in (\mathbb{R}^{N_x})^{\mathbb{Z}-}$ satisfying

$$\|\overleftarrow{\mathbf{x}}_1 - \overleftarrow{\mathbf{x}}_2\|_w < \delta, \tag{15}$$

we have

$$\|\mathcal{U}(\overleftarrow{\mathbf{x}}_1) - \mathcal{U}(\overleftarrow{\mathbf{x}}_2)\|_w < \epsilon. \tag{16}$$

ESNs have been shown to universally approximate *time-invariant, causal filters with the fading memory property* (Grigoryeva & Ortega, 2018a) (Theorem 4.1).

# D. Proofs

## D.1. Proof of Theorem 4.1 (ESP of ParalESN)

We first prove the sufficient condition. Let $\mathbf{h}_0$ and $\mathbf{h}_0'$ be two vectors in $\mathbb{C}^{N_h}$, let $\mathbf{x}_1, \ldots, \mathbf{x}_N$ be a sequence of inputs, and let $\rho(\bar{\mathbf{\Lambda}}_h) < 1 - \epsilon$ be the spectral radius of $\bar{\mathbf{\Lambda}}_h$. Since $\bar{\mathbf{\Lambda}}_h$ is diagonal, we have $\rho(\bar{\mathbf{\Lambda}}_h) = \|\bar{\mathbf{\Lambda}}_h\|_2$. Then,

$$\begin{aligned}
\|\mathbf{h}_N - \mathbf{h}_N'\|_2 &= \|\bar{\mathbf{\Lambda}}_h \mathbf{h}_{N-1} + \tau(\mathbf{W}_{in}\mathbf{x}_N + \mathbf{b}) - \bar{\mathbf{\Lambda}}_h \mathbf{h}_{N-1}' - \tau(\mathbf{W}_{in}\mathbf{x}_N + \mathbf{b})\|_2 \\
&= \|\bar{\mathbf{\Lambda}}_h(\mathbf{h}_{N-1} - \mathbf{h}_{N-1}')\|_2 \\
&\leq \|\bar{\mathbf{\Lambda}}_h\|_2 \cdot \|(\mathbf{h}_{N-1} - \mathbf{h}_{N-1}')\|_2 \\
&= \rho(\bar{\mathbf{\Lambda}}_h) \cdot \|(\mathbf{h}_{N-1} - \mathbf{h}_{N-1}')\|_2 \\
&\leq (1 - \epsilon) \cdot \|(\mathbf{h}_{N-1} - \mathbf{h}_{N-1}')\|_2 \\
&\vdots \\
&\leq (1 - \epsilon)^N \cdot \|\mathbf{h}_0 - \mathbf{h}_0'\|_2 \xrightarrow{N \to \infty} 0.
\end{aligned}$$

This proves the sufficient condition. To prove the necessary condition, suppose that some diagonal entry $\bar{\lambda}_i$ of $\bar{\mathbf{\Lambda}}_h$ satisfies $|\bar{\lambda}_i| \geq 1$. Given two initial states $\mathbf{h}_0$ and $\mathbf{h}_0'$ such that $(\mathbf{h}_0)_i \neq (\mathbf{h}_0')_i$, unrolling the linear recurrence yields the explicit formulas for $\mathbf{h}_N$ and $\mathbf{h}_N'$:

$$\mathbf{h}_N = \bar{\mathbf{\Lambda}}_h^N \mathbf{h}_0 + \sum_{j=1}^N \bar{\mathbf{\Lambda}}_h^{N-j} \tau(\mathbf{W}_{in}\mathbf{x}_j + \mathbf{b}) \tag{17}$$

$$\mathbf{h}_N' = \bar{\mathbf{\Lambda}}_h^N \mathbf{h}_0' + \sum_{j=1}^N \bar{\mathbf{\Lambda}}_h^{N-j} \tau(\mathbf{W}_{in}\mathbf{x}_j + \mathbf{b}) \tag{18}$$

Subtracting these two equations gives

$$\mathbf{h}_N - \mathbf{h}_N' = \bar{\mathbf{\Lambda}}_h^N(\mathbf{h}_0 - \mathbf{h}_0'). \tag{19}$$

Focusing on component $i$, we obtain $(\mathbf{h}_N)_i - (\mathbf{h}_N')_i = \bar{\lambda}_i^N((\mathbf{h}_0)_i - (\mathbf{h}_0')_i)$. Since $(\mathbf{h}_0')_i \neq (\mathbf{h}_0)_i$ and $|\bar{\lambda}_i| \geq 1$, this term never vanishes: it grows exponentially in magnitude if $|\lambda_i| > 1$, and remains equal to $(\mathbf{h}_0')_i - (\mathbf{h}_0)_i$ if $\lambda_i = 1$.

## D.2. Proof of Proposition 4.2

Since $\mathbf{W}_h$ is diagonalizable, we can write $\mathbf{W}_h = \mathbf{V}\mathbf{\Lambda}_h\mathbf{V}^{-1}$, with $\mathbf{V} \in \mathbb{C}^{N_h \times N_h}$ and $\mathbf{\Lambda}_h$ a diagonal matrix whose diagonal entries are the eigenvalues of $\mathbf{W}_h$. We can rewrite the recurrence in terms of $\mathbf{V}$ and $\mathbf{\Lambda}$ as

$$\mathbf{h}_t = \mathbf{V}\mathbf{\Lambda}_h\mathbf{V}^{-1}\mathbf{h}_{t-1} + \mathbf{W}_{in}\mathbf{x}_t. \tag{20}$$

Pre-multiplying both sides of (20) by $\mathbf{V}^{-1}$ yields a complex diagonal ESN with equations

$$
\begin{cases}
\tilde{\mathbf{h}}_\mathrm{t} = \boldsymbol{\Lambda}_\mathrm{h}\tilde{\mathbf{h}}_{t-1} + \tilde{\mathbf{W}}_\mathrm{in}\mathbf{x}_\mathrm{t} \\
\mathbf{y}_\mathrm{t} = \mathbf{W}_\mathrm{out}(\sigma(\tilde{\mathbf{W}}_\mathrm{h}\tilde{\mathbf{h}}_\mathrm{t}))
\end{cases}
$$

where $\tilde{\mathbf{h}}_\mathrm{t} = \mathbf{V}^{-1}\mathbf{h}_\mathrm{t}$, $\tilde{\mathbf{W}}_\mathrm{in} = \mathbf{V}^{-1}\mathbf{W}_\mathrm{in}$, and $\tilde{\mathbf{W}}_\mathrm{h} = \mathbf{W}_\mathrm{h}\mathbf{V}$. Therefore, for any linear ESN and any input sequence, there exists an equivalent complex diagonal ESN that, given the same inputs, produces identical outputs.

### D.3. Proof of Theorem 4.3

Diagonalizable matrices are dense in $\mathbb{C}^{N \times N}$; therefore, every matrix is diagonalizable up to an arbitrarily small perturbation (Hartfiel, 1995). That is, given any matrix $\mathbf{W}_\mathrm{h}$, there exists $\mathbf{W}_\mathrm{h}^\delta$ such that

$$
\|\mathbf{W}_\mathrm{h} - \mathbf{W}_\mathrm{h}^\delta\| < \delta, \tag{21}
$$

and $\mathbf{W}_\mathrm{h}^\delta$ is diagonalizable.

Since we introduce a small error term $\delta$, we must rule out the possibility that this error grows over an infinite sequence of inputs (as in the settings of (Grigoryeva & Ortega, 2018b; Gonon & Ortega, 2020)). To this end, we introduce a lemma establishing that the error in the induced filter can indeed be controlled, provided the perturbation is sufficiently small.

**Lemma D.1.** *Let $\mathcal{E}$ be a filter induced by a linear ESN with transition matrix $\mathbf{W}_h$ satisfying $\rho(\mathbf{W}_h) < 1$ (i.e., the linear ESN has the ESP). Then for any such $\mathcal{E}$, any $\epsilon > 0$, and any compact set $K \subset \mathbb{R}^{N_{in}}$, there exists $\delta > 0$ such that for any filter $\mathcal{E}^\delta$ induced by a transition matrix $\mathbf{W}_h^\delta$ with $\|\mathbf{W}_h - \mathbf{W}_h^\delta\| < \delta$ (where $\|\cdot\|$ denotes a matrix norm), we have $\|\mathcal{E} - \mathcal{E}^\delta\|_\infty = \sup_{x \in K^{\mathbb{Z}_-}} |\mathcal{E}(x) - \mathcal{E}^\delta(x)| < \epsilon$.*

*Proof.* We aim to show that $\|\mathcal{E} - \mathcal{E}^\delta\|_\infty = \sup_{x \in K^{\mathbb{Z}_-}} |\mathcal{E}(x) - \mathcal{E}^\delta(x)| < \epsilon$. Let $M$ be a constant such that $|\mathbf{x}_\mathrm{t}| < M$ for all $\mathbf{x}_\mathrm{t} \in K$. Then, for any fixed $t$,

$$
\mathbf{h}_\mathrm{t} - \mathbf{h}_\mathrm{t}^\delta = \sum_{k=1}^{\infty} \mathbf{W}_\mathrm{h}^k \mathbf{W}_\mathrm{in}\mathbf{x}_{t-k} - \sum_{k=1}^{\infty} (\mathbf{W}_\mathrm{h}^\delta)^k \mathbf{W}_\mathrm{in}\mathbf{x}_{t-k} \tag{22}
$$

$$
= \sum_{k=0}^{\infty} (\mathbf{W}_\mathrm{h}^k - (\mathbf{W}_\mathrm{h}^\delta)^k)\mathbf{W}_\mathrm{in}\mathbf{x}_{t-k}. \tag{23}
$$

Taking norms,

$$
\|\mathbf{h}_\mathrm{t} - \mathbf{h}_\mathrm{t}^\delta\| = \left\|\sum_{k=0}^{\infty} (\mathbf{W}_\mathrm{h}^k - (\mathbf{W}_\mathrm{h}^\delta)^k)\mathbf{W}_\mathrm{in}\mathbf{x}_{t-k}\right\| \tag{24}
$$

$$
\leq \sum_{k} \|\mathbf{W}_\mathrm{in}\||\mathbf{x}_{t-k}| \cdot \|(\mathbf{W}_\mathrm{h}^k - (\mathbf{W}_\mathrm{h}^\delta)^k)\| \tag{25}
$$

$$
\leq M' \sum_{k} \|(\mathbf{W}_\mathrm{h}^k - (\mathbf{W}_\mathrm{h}^\delta)^k)\| \tag{26}
$$

where $M' = M\|\mathbf{W}_\mathrm{in}\|$. We now apply the telescoping identity to expand the difference of matrix powers:

$$
\|(\mathbf{W}_\mathrm{h}^k - (\mathbf{W}_\mathrm{h}^\delta)^k)\| = \left\|\sum_{j=0}^{k-1} \mathbf{W}_\mathrm{h}^j(\mathbf{W}_\mathrm{h} - \mathbf{W}_\mathrm{h}^\delta)(\mathbf{W}_\mathrm{h}^\delta)^{k-1-j}\right\| \tag{27}
$$

$$
\leq \sum_{j=0}^{k-1} \|\mathbf{W}_\mathrm{h}^j\| \cdot \|\mathbf{W}_\mathrm{h} - \mathbf{W}_\mathrm{h}^\delta\| \cdot \|(\mathbf{W}_\mathrm{h}^\delta)^{k-1-j}\|. \tag{28}
$$

It is a standard result in linear algebra that if $\rho(\mathbf{W}) < 1$, then there exist constants $C(\mathbf{W}) > 0$ and $\alpha(\mathbf{W})$ with $\rho(\mathbf{W}) < \alpha(\mathbf{W}) < 1$ such that $\|\mathbf{W}^k\| \leq C\alpha^k$ (see e.g. (Horn & Johnson, 1990), Corollary 5.6.13). Setting

$C = \max\{C(\mathbf{W}_{\mathrm{h}}), C(\mathbf{W}_h^\delta)\}$ and $\alpha = \max\{\alpha(\mathbf{W}_{\mathrm{h}}), \alpha(\mathbf{W}_h^\delta)\}$, we obtain

$$\left\|(\mathbf{W}_{\mathrm{h}}^k - (\mathbf{W}_h^\delta)^k)\right\| \leq \sum_{j=0}^{k-1} C\alpha^j \delta C\alpha^{k-1-j} = kC^2\alpha^{k-1}\delta.$$

Summing over all $k$ gives, for every $t$,

$$\|\mathbf{h}_{\mathrm{t}} - \mathbf{h}_{\mathrm{t}}^\delta\| \leq M'C^2\delta \sum_k k\alpha^{k-1} = M'C^2\delta S,$$

where $S = \sum_k k\alpha^{k-1} = \alpha/(1-\alpha)^2$ is the derivative of the geometric series with ratio $\alpha < 1$, and is therefore finite. Choosing $\delta < \frac{\epsilon}{M'C^2 S}$ yields the desired bound. $\square$

*Remark* D.2. Lemma D.1 extends naturally to a stochastic setting. Let $(\mathbf{x}_{\mathrm{t}})_{t \in \mathbb{Z}_-}$ be a stochastic process defined on a probability space $(\Omega, \mathcal{F}, \mu)$, and assume that:

- $(\mathbf{x}_{\mathrm{t}})$ is *stationary*, i.e., for any finite collection of indices $t_1, \ldots, t_k$ and any shift $\tau \in \mathbb{Z}_-$, the marginal distributions satisfy
$$p(\mathbf{x}_{t_1}, \ldots, \mathbf{x}_{t_k}) = p(\mathbf{x}_{t_1+\tau}, \ldots, \mathbf{x}_{t_k+\tau});$$

- $\mathbf{x}_0 \in L^p(\mu)$ for some $1 \leq p < \infty$, i.e.,
$$\int |\mathbf{x}_0|^p \, d\mu(\mathbf{x}_0) < \infty.$$

By stationarity,
$$\|\mathbf{x}_{\mathrm{t}}\|_{L^p} = \|\mathbf{x}_0\|_{L^p} < \infty \quad \text{for all } t \in \mathbb{Z}_-,$$

so the input process is uniformly bounded in $L^p$. Under these assumptions, Lemma D.1 can be proved via the same chain of inequalities, and the perturbation bound therefore holds in $L^p(\mu)$. Note that these assumptions are strictly weaker than those required by the universality results of (Gonon & Ortega, 2020): the latter impose stronger integrability conditions, whereas the present lemma requires only $L^p$-boundedness and stationarity. The theorems of (Gonon & Ortega, 2020) therefore apply.

Finally, we prove the theorem. Since linear ESNs are universal in the class of fading memory filters (Grigoryeva & Ortega, 2018b; Gonon & Ortega, 2020), it suffices to show that ParalESN is dense in the set of linear ESNs with the ESP. Since the set of non-diagonalizable matrices over $\mathbb{C}$ has Lebesgue measure zero, given any linear ESN filter $\mathcal{E}$ and any $\epsilon > 0$, we can perturb its transition matrix $\mathbf{W}_{\mathrm{h}}$ by $\delta$ small enough so that $\mathbf{W}_h^\delta$ satisfies the conditions of Lemma D.1 and is diagonalizable. The resulting filter $\mathcal{E}'$ is then exactly equivalent to a ParalESN filter $\mathcal{D}$ with transition matrix $\mathbf{\Lambda}$ given by the eigendecomposition of $\mathbf{W}_h^\delta$, and $\|\mathcal{E} - \mathcal{D}\| \leq \epsilon$.

# E. Datasets

## E.1. Memory-based

**MemCap.** The MemCap task consists of outputting a delayed version of the input $k$ time steps in the past. The memory capacity score, used to assess performance on this task, is computed by summing the squared correlation coefficients between the output and the $k$-step delayed input, for each delay $k = 1, \ldots, 200$. We generate an input sequence drawn uniformly from $[-0.8, 0.8]$ with length $T = 7000$. The first 5000 time steps are used for training, the next 1000 for validation, and the final 1000 for testing. Both training and inference employ a 100 time step washout to warm up the reservoir.

**ctXOR.** Consider a one-dimensional input time series $\mathbf{x}(t)$ drawn uniformly from $(-0.8, 0.8)$, and let $\mathbf{r}(t-d) = \mathbf{x}(t-d-1)\mathbf{x}(t-d)$. The task is to output $\mathbf{y}(t) = \mathbf{r}(t-d)^2 \operatorname{sign}(\mathbf{r}(t-d))$, where $d$ is the delay. We consider delays $d = 5$ (ctXOR5) and $d = 10$ (ctXOR10).

**SinMem.** Given a one-dimensional input time series $\mathbf{x}(t)$ drawn uniformly from $(-0.8, 0.8)$, the task is to output $\mathbf{y}(t) = \sin(\pi\mathbf{x}(t-d))$. We consider delays $d = 10$ (SinMem10) and $d = 20$ (SinMem20).

## E.2. Forecasting

**Lorenz96.** The Lorenz96 task is to predict the next state of the time series $\mathbf{x}(t)$, governed by the following 5-dimensional chaotic system:

$$\frac{\partial f_i}{\partial t}(t) = f_{i-1}(t)(f_{i+1}(t) - f_{i-2}(t)) - f_i(t) + 8, \tag{29}$$

for $i = 1, \ldots, 5$. We focus on predicting the 25th (Lz25) and 50th (Lz50) future states of the time series, i.e., the tasks involve predicting $\mathbf{y}(t) = \mathbf{x}(t + 25)$ and $\mathbf{y}(t) = \mathbf{x}(t + 50)$, respectively. We generate a time series of length $T = 1200$, with the first 400 time steps used for training, the next 400 for validation, and the final 400 for testing.

**Mackey-Glass.** The Mackey-Glass 17 (MG) task is to predict the next state of the following time series:

$$\frac{\partial f}{\partial t}(t) = \frac{0.2f(t - 17)}{1 + f(t - 17)^{10}} - 0.1f(t). \tag{30}$$

We focus on predicting the 1st and 84th future states of the time series, i.e., the tasks involve predicting $\mathbf{y}(t) = \mathbf{x}(t + 1)$ (MG) and $\mathbf{y}(t) = \mathbf{x}(t + 84)$ (MG84), respectively. We generate a time series of length $T = 10000$, with the first 5000 time steps used for training, the next 2500 for validation, and the final 2500 for testing.

**NARMA.** Given a one-dimensional input time series $\mathbf{x}(t)$ drawn uniformly from $[0, 0.5]$, the NARMA task is to predict the next state of the following time series:

$$\mathbf{y}(t) = 0.3\mathbf{y}(t - 1) + 0.01\mathbf{y}(t - 1)\sum_{i=1}^{d}\mathbf{y}(t - i) + 1.5\mathbf{x}(t - d)\mathbf{x}(t - 1) + 0.1. \tag{31}$$

We consider NARMA10 (N10) and NARMA30 (N30), with look-ahead delays of $d = 10$ and $d = 30$, respectively. We generate a time series of length $T = 10000$, with the first 5000 time steps used for training, the next 2500 for validation, and the final 2500 for testing.

**Real-world datasets.** The ETTh1, ETTh2, ETTm1, and ETTm2 datasets (Zhou et al., 2021) represent challenging real-world forecasting tasks. Table 6 provides an overview of their characteristics. We focus on the multivariate case and predict all available features, using a prediction horizon of 192 time steps. Each feature in the training set is independently normalized to zero mean and unit variance; the resulting normalization coefficients are then applied to the validation and test sets. Since order-of-magnitude outliers are present in the training set, the normalized training data is clipped to the interval $(-10, 10)$ across all datasets. The validation and test data remain unmodified.

*Table 6.* Overview of the real-world forecasting datasets. Training, validation, and test sizes are reported in number of time steps.

| DATASET | TRAIN | VALIDATION | TEST | # FEATURES |
|---------|-------|------------|------|------------|
| ETTH1/2 | 8640 | 2880 | 2880 | 7 |
| ETTM1/2 | 34560 | 11520 | 11520 | 7 |

## E.3. Classification

Table 7 provides an overview of the classification datasets. The validation set is obtained via a 90–10 stratified split for sMNIST and psMNIST, and via a 70–30 stratified split for the remaining tasks. No data augmentation is applied.

*Table 7.* Overview of time series and 1-D pixel-level classification datasets.

| DATASET | TRAIN | TEST | LENGTH | # FEATURES | # CLASSES |
|---------|-------|------|--------|------------|-----------|
| BLINK | 500 | 450 | 510 | 4 | 2 |
| FAULTDETECTIONA | 10912 | 2728 | 5120 | 1 | 3 |
| FORDA | 3601 | 1320 | 500 | 1 | 2 |
| FORDB | 3636 | 810 | 500 | 1 | 2 |
| SMNIST/PSMNIST | 60000 | 10000 | 784 | 1 | 10 |
| STARLIGHTCURVES | 1000 | 8236 | 1024 | 1 | 3 |

### E.4. Long Range Arena

The Long Range Arena (LRA) benchmarks are specifically designed to evaluate a model's ability to capture long-range dependencies across diverse data modalities, including images and text. Table 8 provides an overview of the considered datasets.

*Table 8.* Overview of Long Range Arena (LRA) classification datasets.

| DATASET | TRAIN | TEST | LENGTH | # FEATURES | # CLASSES |
|---|---|---|---|---|---|
| IMAGE | 45000 | 10000 | 1024 | 1 | 10 |
| LISTOPS | 96000 | 2000 | 2048 | 1 | 10 |
| TEXT | 25000 | 25000 | 4096 | 1 | 2 |
| PATHFINDER | 160000 | 20000 | 1024 | 1 | 2 |

## F. Experimental Setting

### F.1. Computational Resources

Experiments are run on a system with $4 \times$ 18-core Intel Xeon Gold 6140M CPUs @ 2.30 GHz (144 threads total) and $1 \times$ NVIDIA Tesla V100-PCIE-16GB GPUs. To track computational efficiency metrics such as training time, energy consumption, and $CO_2$ emissions, we use CodeCarbon (Courty et al., 2024).

### F.2. Fully-Trainable Sequence Models

In our experiments on sMNIST and psMNIST, we cover a wide range of fully-trainable sequence models, including recurrent (LSTMs (Hochreiter & Schmidhuber, 1997)), attention-based (Transformers (Vaswani et al., 2017)), and deep state space models (S4[9] (Gu et al., 2022) and LRU[10] (Orvieto et al., 2023), and Mamba[11] (Gu & Dao, 2024)). We run our own training runs to track efficiency metrics such as training time and energy consumption. See ppendix F.3 for details on their hyperparameters and model selection.

For LRA, we also consider various attention-based models, S5(Smith et al., 2023), and RWKV (Peng et al., 2023). We report results from the corresponding papers.

### F.3. Model Selection

Model selection is performed via Bayesian search, exploring configurations up to a maximum runtime of 24 hours. The best configuration is chosen based on validation set performance.

For RC models, Table 9 provides an overview of the hyperparameter values explored. Common hyperparameters for both ESN and ParalESN include the bias scaling $\omega_b$, the leaky rate $\tau$, and the number of layers $L$. For traditional RC, we additionally explore the spectral radius $\rho$, used to rescale the recurrent weight matrix, and the input scaling $\omega_{in}$, used to scale the input weight matrix. For ParalESN, we explore the minimum and maximum magnitudes of the diagonal entries, $\rho_{min}$ and $\rho_{max}$, and the minimum and maximum phases (i.e., the angle with the positive $x$-axis), $\theta_{min}$ and $\theta_{max}$, which control the eigenvalues of the diagonal transition matrix. We additionally explore $\omega_{mix}$ and $\omega_{mixb}$ for scaling the kernel and bias vector of the mixing layer, and $k$ for the kernel size. For the deep configurations, we explore the number of (untrained) reservoir layers $L$ and an additional set of (inter) hyperparameters for layers beyond the first, to

*Table 9.* Hyperparameters for RC models.

| HYPERPARAMETERS | VALUES |
|---|---|
| `concat` | [False, True] |
| $L$ | [1, 2, 3, 4, 5] |
| $\omega_b$ and inter-$\omega_b$ | $\{0, 0.01, 0.1, 1, 10\}$ |
| $\tau$ and inter-$\tau$ | $\{0.1, 0.5, 0.9, 1\}$ |
| **(ESN)** | |
| $\omega_{in}$ and inter-$\omega_x$ | $\{0.01, 0.1, 1, 10\}$ |
| $\rho$ and inter-$\rho$ | $\{0.1, 0.5, 0.9\}$ |
| **(ParalESN)** | |
| *(Reservoir)* | |
| $\rho_{max}$ and inter-$\rho_{max}$ | $\{0.1, 0.5, 0.9\}$ |
| $\rho_{min}$ and inter-$\rho_{min}$ | $\{0, 0.1, 0.5, 0.9\}$ |
| $\theta_{max}$ and inter-$\theta_{max}$ | $\{\frac{1}{2}\pi, \pi, 2\pi\}$ |
| $\theta_{min}$ and inter-$\theta_{min}$ | $\{0, \frac{1}{2}\pi, \pi, 2\pi\}$ |
| *(Mixer)* | |
| $\omega_{mix}$ and inter-$\omega_{mix}$ | $\{0.01, 0.1, 1, 10\}$ |
| $\omega_{mixb}$ and inter-$\omega_{mixb}$ | $\{0, 0.01, 0.1, 1, 10\}$ |
| $k$ and inter-$k$ | $\{3, 5, 7, 9\}$ |

---

[9]Implementation from https://github.com/state-spaces/s4.

[10]Implementation from github.com/NicolasZucchet/minimal-LRU.

[11]Implementation from github.com/state-spaces/mamba.

promote diverse dynamics across layers. We also consider the hyperparameter `concat` $\in$ {False, True}, which determines whether the readout receives the states of the last layer only or the concatenation of states across all layers. To ensure a consistent number of trainable parameters when states are concatenated, the total number of reservoir units is evenly divided across layers; any remainder is allocated to the first layer. When the readout is implemented as a ridge regressor, we explore regularization strengths in {0, 0.01, 0.1, 1, 10, 100}. When implemented as an MLP, it is trained with the Adam optimizer at a learning rate of $5 \times 10^{-4}$ with all other settings left at their PyTorch defaults. The readout is implemented as a ridge regressor trained via a Singular Value Decomposition (SVD) solver for memory-based, forecasting, and time series classification tasks, and as a 2-layer Multi-Layer Perceptron (MLP) for the MNIST benchmarks. Prior to the readout, the reservoir states are always standardized to zero mean and unit variance.

For fully-trainable models on sMNIST and psMNIST, we use the Adam optimizer and sweep the learning rate with a log-uniform distribution over $[0.0001, 0.01]$ and the weight decay over $\{0, 10^{-7}, 10^{-6}, 10^{-5}, 10^{-4}, 10^{-3}, 10^{-2}, 10^{-1}\}$. For LRU, we additionally sweep $\rho_{\min}$ with a uniform distribution over $[0, 1]$, $\rho_{\max}$ over $[0.8, 1]$, and $\theta_{\max}$ over $[0.001, \pi]$. The LSTM uses a hidden size of 192. The Transformer uses an input size of 96 and a feedforward size of 128 with 3 encoder layers. LRU uses 3 layers with a hidden size of 82. Mamba uses 6 layers with model dimension 64, state expansion factor 16, and local convolutional width 4. S4 uses 2 layers with a hidden size of 172. Fully-trainable models and the MLP readout of ParalESN are trained for up to 100 epochs with early stopping, to avoid inflating the computational efficiency metrics. For classification tasks, the classification head of each model receives only the hidden state at the final time step.

## G. Additional Experiments

### G.1. Hyperparameter Sensitivity

Figures 6 and 7 analyze the hyperparameter sensitivity of ParalESN and ParalESN (deep) on NARMA10 (N10). Recall that, for ParalESN (deep), we use the prefix *inter-* to denote the hyperparameters of all layers beyond the first, while the unprefixed notation is used for those of the first layer. As is common in RC, both models are sensitive to most of their hyperparameters in both the shallow and deep configurations, a consequence of relying on untrained dynamics. The most critical hyperparameters are consistent across both configurations (e.g., spectral radii and phases), while the kernel size has the smallest impact. Although careful model selection is essential for predictive performance, we note that ParalESN's speed advantage over traditional RC makes it possible to evaluate significantly more configurations within the same time budget, which partially mitigates this concern.

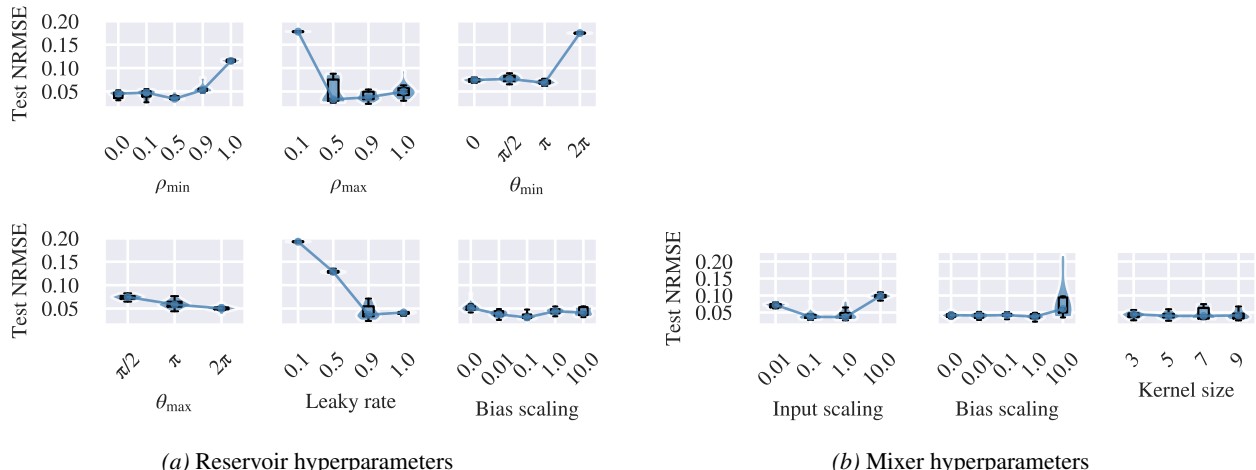

*(a)* Reservoir hyperparameters          *(b)* Mixer hyperparameters

*Figure 6.* Sensitivity of ParalESN to its hyperparameters for *(a)* the reservoir layer and *(b)* the mixing layer, on the N10 task.

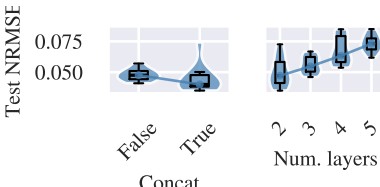

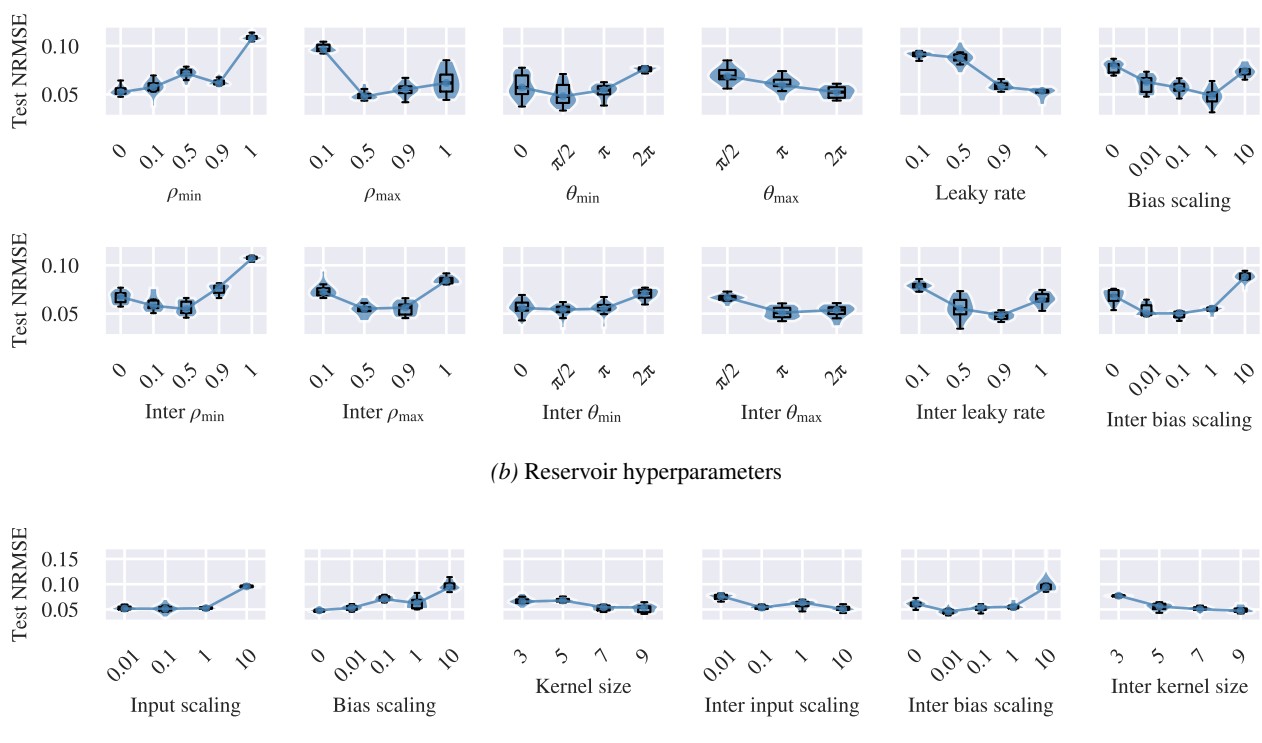

*(a)* Number of layers and concatenation

*(b)* Reservoir hyperparameters

*(c)* Mixer hyperparameters

**Figure 7.** Sensitivity of ParalESN (deep) to its hyperparameters for *(a)* number of layers and concatenation, *(b)* the reservoir layer, and *(c)* the mixing layer, on the N10 task.

### G.2. Parameter Efficiency

As highlighted by the computational complexity analysis in Appendix B and Table 5, ParalESN is significantly more parameter-efficient than traditional RC. This efficiency stems from the use of diagonal transition matrices in place of dense matrices. Table 10 provides an empirical comparison of the total parameter counts of ParalESN and traditional ESNs, excluding the readout layer, whose parameters are identical across all models. For the same number of recurrent neurons, ParalESN has orders of magnitude fewer parameters.

*Table 10.* Total number of parameters in the recurrence of ParalESN and traditional ESNs, assuming input dimension $N_{in} = 1$, 1024 recurrent neurons per layer, and 5 layers for deep configurations.

| MODEL | TOTAL PARAMS. |
|---|---|
| ESN | $\approx 1.1M$ |
| ESN (deep) | $\approx 9.4M$ |
| ParalESN | $\approx 2.1K$ |
| ParalESN (deep) | $\approx 10.2K$ |

### G.3. Complex vs. Real-Valued Parameterization

Whether complex-valued representations are strictly necessary compared to real-valued alternatives is an active area of investigation. Recent work has shown that complex-valued parameterizations offer benefits both in terms of training dynamics and long-term memory retention (Orvieto et al., 2024). Conceptually, complex eigenvalues enable oscillatory dynamics, allowing the model to capture periodic and quasi-periodic patterns in the input signal — a property that is

particularly relevant for time series processing.

Motivated by these insights, ParalESN employs complex-valued parameterization. We validate this design choice by comparing (complex-valued) ParalESN against a corresponding real-valued variant, which we term *rParalESN*. Tables 11 and 12 present results on memory-based and forecasting tasks, respectively. Complex-valued ParalESN consistently outperforms rParalESN on the majority of tasks, sometimes by an order of magnitude.

*Table 11.* Comparison between complex-valued and real-valued ParalESN (*rParalESN*) on memory-based tasks, assuming 128 recurrent neurons for each model. The **best result** is highlighted in bold.

| MEMORY-BASED | $\uparrow$ MEMCAP | $\cdot 10^{-1}$ $\downarrow$ CTXOR5 | $\cdot 10^{-1}$ $\downarrow$ CTXOR10 | $\cdot 10^{-1}$ $\downarrow$ SINMEM10 | $\cdot 10^{-1}$ $\downarrow$ SINMEM20 |
|---|---|---|---|---|---|
| rParalESN | $27.8_{\pm 0.3}$ | $8.4_{\pm 0.2}$ | $10.1_{\pm 0.1}$ | $4.3_{\pm 0.1}$ | $8.7_{\pm 0.1}$ |
| rParalESN (deep) | $29.7_{\pm 0.6}$ | $4.7_{\pm 0.6}$ | $9.2_{\pm 0.1}$ | $6.7_{\pm 0.3}$ | $8.9_{\pm 0.1}$ |
| ParalESN | $114.5_{\pm 1.4}$ | $3.9_{\pm 0.1}$ | $8.2_{\pm 0.2}$ | $3.7_{\pm 0.0}$ | $3.7_{\pm 0.0}$ |
| ParalESN (deep) | $\mathbf{125.0_{\pm 0.2}}$ | $\mathbf{3.6_{\pm 0.1}}$ | $\mathbf{5.6_{\pm 0.4}}$ | $\mathbf{1.0_{\pm 0.2}}$ | $\mathbf{2.5_{\pm 0.4}}$ |

*Table 12.* Comparison between complex-valued and real-valued ParalESN (*rParalESN*) on forecasting tasks, assuming 128 recurrent neurons for each model. The **best result** is highlighted in bold.

| FORECASTING | $\cdot 10^{-2}$ $\downarrow$ Lz25 | $\cdot 10^{-2}$ $\downarrow$ Lz50 | $\cdot 10^{-4}$ $\downarrow$ MG | $\cdot 10^{-2}$ $\downarrow$ MG84 | $\cdot 10^{-2}$ $\downarrow$ N10 | $\cdot 10^{-2}$ $\downarrow$ N30 | $\cdot 10^{-1}$ $\downarrow$ ETTH1 | $\cdot 10^{-1}$ $\downarrow$ ETTH2 | $\cdot 10^{-1}$ $\downarrow$ ETTM1 | $\cdot 10^{-1}$ $\downarrow$ ETTM2 |
|---|---|---|---|---|---|---|---|---|---|---|
| rParalESN | $11.2_{\pm 0.4}$ | $\mathbf{29.3_{\pm 0.6}}$ | $3.8_{\pm 0.2}$ | $9.9_{\pm 1.1}$ | $10.5_{\pm 0.2}$ | $17.8_{\pm 0.1}$ | $\mathbf{8.8_{\pm 0.1}}$ | $11.3_{\pm 0.9}$ | $6.6_{\pm 0.1}$ | $5.2_{\pm 0.2}$ |
| rParalESN (deep) | $11.2_{\pm 0.4}$ | $31.6_{\pm 0.5}$ | $3.7_{\pm 0.2}$ | $8.8_{\pm 0.7}$ | $11.8_{\pm 0.6}$ | $18.5_{\pm 0.1}$ | $9.0_{\pm 0.1}$ | $10.8_{\pm 0.7}$ | $6.6_{\pm 0.1}$ | $\mathbf{4.9_{\pm 0.1}}$ |
| ParalESN | $10.4_{\pm 0.5}$ | $30.2_{\pm 0.5}$ | $2.8_{\pm 0.2}$ | $7.4_{\pm 0.4}$ | $\mathbf{3.7_{\pm 0.8}}$ | $10.2_{\pm 0.1}$ | $9.0_{\pm 0.2}$ | $12.8_{\pm 1.2}$ | $\mathbf{6.5_{\pm 0.1}}$ | $5.1_{\pm 0.1}$ |
| ParalESN (deep) | $\mathbf{10.3_{\pm 0.3}}$ | $29.4_{\pm 0.3}$ | $\mathbf{2.6_{\pm 0.4}}$ | $\mathbf{5.2_{\pm 0.5}}$ | $4.5_{\pm 1.0}$ | $10.2_{\pm 0.2}$ | $8.8_{\pm 0.1}$ | $\mathbf{9.7_{\pm 0.9}}$ | $\mathbf{6.5_{\pm 0.0}}$ | $5.0_{\pm 0.0}$ |

## G.4. Comparison with Structured Reservoir Computing

We compare ParalESN with other structured reservoir computing approaches, including the Simple Cycle Reservoir (SCR) (Rodan & Tino, 2011) and Structured Reservoir Computing (Structured RC) (Dong et al., 2020)[12]. For both SCR and Structured RC, model selection follows the methodology described in Appendix F. Note that SCR does not require tuning the spectral radius, since the transition matrix employs a fixed ring topology. For Structured RC, we tune the reservoir scaling over $\{0.01, 0.1, 1, 10\}$ rather than the spectral radius. Although traditional ESNs are not based on structured transforms, their results are included as a baseline.

Table 13 presents the results for a shallow, single-layer configuration of each model. ParalESN is consistently among the top-performing models alongside ESN, while other structured transform approaches consistently underperform across all datasets. In particular, compared to SCR and Structured RC, ParalESN achieves lower test error on MG, MG84, and N10 by an entire order of magnitude, and roughly half the error on N30.

*Table 13.* Comparison of structured reservoir computing approaches on time series forecasting, assuming 128 recurrent neurons for each model. A traditional ESN is included as a baseline. The **best result** is highlighted in bold; second-best is underlined.

| FORECASTING | $\cdot 10^{-2}$ $\downarrow$ Lz25 | $\cdot 10^{-2}$ $\downarrow$ Lz50 | $\cdot 10^{-4}$ $\downarrow$ MG | $\cdot 10^{-2}$ $\downarrow$ MG84 | $\cdot 10^{-2}$ $\downarrow$ N10 | $\cdot 10^{-2}$ $\downarrow$ N30 |
|---|---|---|---|---|---|---|
| ESN | $\mathbf{10.0_{\pm 0.3}}$ | $30.8_{\pm 0.6}$ | $3.0_{\pm 0.0}$ | $\mathbf{6.5_{\pm 0.4}}$ | $\mathbf{2.7_{\pm 0.4}}$ | $10.3_{\pm 0.1}$ |
| SCR | $22.1_{\pm 0.4}$ | $35.6_{\pm 0.8}$ | $18.7_{\pm 4.2}$ | $32.2_{\pm 3.1}$ | $11.1_{\pm 0.2}$ | $16.1_{\pm 0.9}$ |
| Structured RC | $10.4_{\pm 0.2}$ | $30.9_{\pm 0.5}$ | $12.6_{\pm 0.1}$ | $28.5_{\pm 1.6}$ | $18.7_{\pm 0.1}$ | $20.9_{\pm 0.1}$ |
| ParalESN | $10.4_{\pm 0.5}$ | $\mathbf{30.2_{\pm 0.5}}$ | $\mathbf{2.8_{\pm 0.2}}$ | $7.4_{\pm 0.4}$ | $3.7_{\pm 0.8}$ | $\mathbf{10.2_{\pm 0.1}}$ |

## G.5. Long-Range Temporal Modeling

To evaluate the long-range temporal modeling capabilities of ParalESN, we consider a selection of Long Range Arena (LRA) benchmarks (Tay et al., 2021), including Image, ListOps, Text, and PathFinder.

---

[12]Implementation from github.com/rubenohana/Reservoir-computing-kernels.

**Discussion.** Table 14 shows that ParalESN (deep) consistently outperforms Transformer-based baselines across all benchmarks. While it does not consistently match the performance of SSM-based and RNN-based sequence models, ParalESN (deep) achieves competitive performance on the Text task and is broadly on par with RWKV overall, despite relying on untrained reservoir dynamics and requiring no backpropagation through time.

*Table 14.* Test set results on LRA benchmarks. The **best result** is highlighted in bold.

| LONG RANGE ARENA | ↑ IMAGE | ↑ LISTOPS | ↑ TEXT | ↑ PATHFINDER | ↑ AVG |
|---|---|---|---|---|---|
| RANDOM | 10.00 | 10.00 | 50.00 | 50.00 | 30.00 |
| (*Transformer-based*) | | | | | |
| Transformer | 42.4 | 36.4 | 64.3 | 71.4 | 53.6 |
| Reformer | 38.1 | 37.3 | 56.1 | 68.5 | 50.0 |
| BigBird | 40.8 | 36.1 | 64.0 | 74.9 | 53.9 |
| Linear Trans. | 42.3 | 16.1 | 65.9 | 75.3 | 49.9 |
| Performer | 42.8 | 18.0 | 65.4 | 77.1 | 50.8 |
| (*SSM-based*) | | | | | |
| S4 | **88.7** | 59.6 | 86.8 | 94.2 | 82.3 |
| S5 | 88.0 | **62.2** | 89.3 | **95.3** | **84.7** |
| LRU | - | 60.2 | **89.4** | 95.1 | 81.6 |
| (*RNN-based*) | | | | | |
| RWKV | 70.5 | 55.9 | 86.0 | 58.4 | 67.7 |
| (*Ours*) | | | | | |
| ParalESN (deep) | 57.2 | 39.6 | 80.2 | 77.7 | 63.7 |

