# OpenReview forum: "ParalESN: Enabling parallel information processing in Reservoir Computing"
_ICML.cc/2026/Conference — ICML 2026 regular_

### Official Review · Reviewer_FuZ2 · 2026-03-04

**Soundness:** 3
**Presentation:** 3
**Significance:** 3
**Originality:** 3
**Overall Recommendation:** 5
**Confidence:** 4

**Summary:**

This paper proposes an efficient version of echo state networks, called ParalESN, which leverages an associative scan enabled by the proposed diagonal linear recurrence, inspired by LRUs. The authors show the efficiency of their method compared to standard ESNs and other fully-trainable recurrent models.

**Compliance With Llm Reviewing Policy:**

Affirmed.

**Final Justification:**

The authors addressed all my questions, the paper has a very good quality and interesting results. Due to the extra experimental insights, I'm raising my score to 5.

**Key Questions For Authors:**

Q1: ParalESN is more parameter-efficient due to its diagonal recurrence. Could you include a comparison of ESN and ParalESN in terms of parameter count on the experimental setups (Table 1,2,3), as those use the same number of recurrent units?

Q2: In Table 4, when the parameter count is matched, what is the exact configuration of the deep ParalESN (number of layers and number of recurrent neurons per layer)?

Q3: What is the influence of the number of layers? Since a deep version is proposed, it would be helpful to analyze the effect of varying the number of layers, especially given that the experiments fix this value to 5.

Q4: Are complex-valued representations actually necessary? What about using real-valued matrices? (Similar to the Real-Gated Linear Recurrent Unit (RG-LRU) variant, which uses real-valued matrices in Soham De et al. Griffin: Mixing Gated Linear Recurrences with Local Attention for Efficient Language Models)

**Limitations:**

Yes

**Strengths And Weaknesses:**

Strengths:
- Well written and well structured
- Clever idea of drawing inspiration from LRU and proposing linear recurrence
- Solid contribution to the field of reservoir computing

Weaknesses:
- Some technical details could be clarified
- Ablation studies/additional small experiments would strengthen the paper
- See questions below

---

> ### Author Rebuttal · Authors · 2026-03-30
>
> Thank you very much for your positive evaluation of our paper. We appreciate that you recognize our work as a solid contribution to the field of reservoir computing.
>
> ---
>
> ### **Q1: On parameter efficiency**
> Including these counts in Tables 1, 2, and 3 would be impractical and would not ease visualization of the results. The optimal configuration of the deep configurations may consist of a different number of (untrained) layers depending on the task. Thus we would need a dedicated parameter count for each task.
>
> To address your concern, we updated the paper with a comparison between the parameter counts of ParalESNs and traditional ESNs. As expected, ParalESN and ParalESN (deep) are orders of magnitude more parameter efficient, mainly due to the fact that they do not employ dense matrices. These parameter counts represent the number of (untrained) parameters in the recurrence and, for simplicity, do not take into account the readout layer as it is the same for both models. The analysis assumes $1024$ recurrent neurons in each layer and $5$ layers for the deep configurations.
>
> | Model | (untrained) Recurrence Params. |
> |---|---|
> | ESN | ≈ 1.1M |
> | ESN (deep) | ≈ 9.4M |
> | ParalESN | ≈ 2.1K |
> | ParalESN (deep) | ≈ 10.2K |
>
> ---
>
> ### **Q2: Configuration of ParalESN (deep) in sMNIST and psMNIST**
> For sMNIST, we have 4 layers with roughly 16K recurrent neurons each. In psMNIST, we have 3 layers with roughly 16K recurrent neurons each.
>
> ---
>
> ### **Q3: On the effect of the number of layers**
> We would like to clarify that the number of layers is not fixed to 5 in the experiments; rather, it is the highest possible value explored during model selection. Thus, while the maximum number of layers we explored is 5, the optimal configuration for ParalESN (deep) may consist of fewer layers depending on the task. We highlight this in Table 8, hyperparameter $L$.
>
> That said, we have updated the hyperparameter sensitivity analysis in the revised paper. Now, we also analyse the effect of the number of layers in the deep version. We observe that adequate tuning of the number of layers is crucial for performance.
>
> **(NARMA10)**
> | Number of layers | Mean Test NRMSE |
> |:----------:|:---------------:|
> | 2          | **0.0450**          |
> | 3          | 0.0559          |
> | 4          | 0.0666          |
> | 5          | 0.0743          |
>
> ---
>
> ### **Q4: On the necessity of complex-valued representations**
> The question of whether complex-valued representations are strictly necessary compared to real-valued alternatives is an active area of investigation in the literature.
>
> From a theoretical standpoint, recent work [R1] shows that complex eigenvalues are not strictly necessary for the expressivity of linear recurrent models. However, the same work demonstrates that complex-valued parameterizations offer benefits both in terms of training dynamics and long-term memory retention. Conceptually, complex eigenvalues allow oscillatory dynamics to emerge, enabling the model to capture periodic and quasi-periodic patterns in the input signal.
>
> Empirically, we conducted additional experiments comparing the proposed complex-valued version with a corresponding real-valued variant, which we term rParalESN. Results suggest that complex-valued representations outperform their real-valued counterparts on the majority of tasks. We believe that a more in-depth investigation is beyond the scope of this work and plan to explore it in future work.
>
> **(Memory-based)**
> | Model | ↑ MemCap | ↓ ctXOR5 | ↓ ctXOR10 | ↓ SinMem10 | ↓ SinMem20 |
> |---|---|---|---|---|---|
> | rParalESN | 27.8 ± 0.3 | 8.4 ± 0.2 | 10.1 ± 0.1 | 4.3 ± 0.1 | 8.7 ± 0.1 |
> | rParalESN (deep) | 29.7 ± 0.6 | 4.7 ± 0.6 | 9.2 ± 0.1 | 6.7 ± 0.3 | 8.9 ± 0.1 |
> | ParalESN | 114.5 ± 1.4 | 3.9 ± 0.1 | 8.2 ± 0.2 | 3.7 ± 0.0 | 3.7 ± 0.0 |
> | ParalESN (deep) | **125.0 ± 0.2** | **3.6 ± 0.1** | **5.6 ± 0.4** | **1.0 ± 0.2** | **2.5 ± 0.4** |
>
> **(Forecasting)**
> | Model | ↓ Lz25 (·10⁻²) | ↓ Lz50 (·10⁻²) | ↓ MG (·10⁻⁴) | ↓ MG84 (·10⁻²) | ↓ N10 (·10⁻²) | ↓ N30 (·10⁻²) | ↓ ETTh1 (·10⁻¹) | ↓ ETTh2 (·10⁻¹) | ↓ ETTm1 (·10⁻¹) | ↓ ETTm2 (·10⁻¹) |
> |---|---|---|---|---|---|---|---|---|---|---|
> | rParalESN | 11.2 ± 0.4 | **29.3 ± 0.6** | 3.8 ± 0.2 | 9.9 ± 1.1 | 10.5 ± 0.2 | 17.8 ± 0.1 | **8.8 ± 0.1** | 11.3 ± 0.9 | 6.6 ± 0.1 | 5.2 ± 0.2 |
> | rParalESN (deep) | 11.2 ± 0.4 | 31.6 ± 0.5 | 3.7 ± 0.2 | 8.8 ± 0.7 | 11.8 ± 0.6 | 18.5 ± 0.1 | 9.0 ± 0.1 | 10.8 ± 0.7 | 6.6 ± 0.1 | **4.9 ± 0.1** |
> | ParalESN | 10.4 ± 0.5 | 30.2 ± 0.5 | 2.8 ± 0.2 | 7.4 ± 0.4 | **3.7 ± 0.8** | **10.2 ± 0.1** | 9.0 ± 0.2 | 12.8 ± 1.2 | **6.5 ± 0.1** | 5.1 ± 0.1 |
> | ParalESN (deep) | **10.3 ± 0.3** | 29.4 ± 0.3 | **2.6 ± 0.4** | **5.2 ± 0.5** | 4.5 ± 1.0 | **10.2 ± 0.2** | **8.8 ± 0.1** | **9.7 ± 0.9** | **6.5 ± 0.0** | 5.0 ± 0.0 |
>
> ---
>
> [R1] A. Orvieto et al. “Universality of linear recurrences followed by non-linear projections: finite-width guarantees and benefits of complex eigenvalues.” International Conference on Machine Learning (2024).

---

> > ### Author Rebuttal · Reviewer_FuZ2 · 2026-04-02
> >
> > I thank the authors for their thorough rebuttal. Good insights and experiments, I recommend adding them to the paper. Even if I had already given a positive score, these further strengthen the paper, so I am raising my score to 5.

---

> > > ### Author Response · Authors · 2026-04-07
> > >
> > > Thank you very much for reading our rebuttal and for raising your score. We are glad that you appreciated the new experiments and insights. We will certainly include them in the revised version of our paper.

---

### Official Review · Reviewer_ewt1 · 2026-03-08

**Soundness:** 2
**Presentation:** 3
**Significance:** 3
**Originality:** 3
**Overall Recommendation:** 4
**Confidence:** 4

**Summary:**

The paper proposes ParalESN, a reservoir computing architecture based on complex diagonal linear recurrence and a fixed nonlinear mixing layer. The main goal is to make the reservoir update parallelizable and more scalable than standard ESNs, while keeping the lightweight training. The paper also gives ESP and expressivity theories and evaluates the method on several benchmarks.

**Compliance With Llm Reviewing Policy:**

Affirmed.

**Final Justification:**

The rebuttal resolves my concerns. I will maintain my positive score 4.

**Key Questions For Authors:**

1. For Proposition 4.2, is the intended claim exact equivalence or approximation in the non-diagonalizable case?

2. Can the authors address the `|lambda_i| = 1` boundary case in Theorem 4.1?

3. Do the reported runtimes include hyperparameter search, or only the final selected run?

4. For the deep variant, do the authors expect the same stability and ESP to hold, and how sensitive is the model to random initialization?

**Limitations:**

The paper does not really discuss the gap between the simplified theoretical analysis and the full architecture.

The paper does not clarify the extent to which the reported efficiency gains include model selection cost.

**Strengths And Weaknesses:**

Strength:

Using a diagonal complex recurrence to remove the sequential bottleneck of ESNs is a sensible and well-motivated direction, and the efficiency gains over standard ESNs are clearly visible in the experiments.

Weakness:

1. Proposition 4.2 seems too strong as stated. The appendix appeals to diagonalizability after arbitrarily small perturbations, but that only supports an approximation argument, not the exact equivalence statement currently written for arbitrary linear reservoirs.

2. Theorem 4.1 is stated as an iff result with `|lambda_i| < 1`, but the appendix necessity argument only explicitly handles `|lambda_i| > 1`, so the boundary case `|lambda_i| = 1` seems missing.

3. The analysis is done for a simplified one-layer formulation, while the practical model in the method section is a bit complex, especially the deep variant. There is no dedicated analysis of sensitivity to random initialization in the deep variant.

4. Since the model has a fairly rich hyperparameter space and uses Bayesian search, I do not think the practical efficiency picture is the same as the runtime test.

---

> ### Author Rebuttal · Authors · 2026-03-30
>
> Thank you very much for the positive evaluation of our paper. We appreciate you highlighting that addressing the sequential bottleneck of traditional ESNs is a sensible and well-motivated direction.
>
> ---
>
> ### **W1 and Q1: Proposition 4.2**
> We agree that Proposition 4.2. is too strong as stated. For this reason, we change the formulation of Proposition 4.2., by adding the hypothesis that $W_{h}$ is diagonalizable. This is not a major issue, since the set of defective (non-diagonalizable) matrices has Lebesgue measure 0 in the space of all square matrices.
>
> This revised formulation requires a more detailed proof of the following results, and in particular of corollary 4.3, in order to account for the possible error introduced by the approximation, although infinitesimal, of a possibly defective matrix with a diagonalizable one. For this reason, we add a further Lemma in Appendix Proofs.
>
> The broad intuition of the new Lemma is that introducing an error $\delta$ in the transition matrix of an ESN, thanks to the ESP / fading memory and the boundedness of the input values, we can still approximate the left-semi-infinite output filter, if $\delta$ is small.
>
> **Lemma.** Let E be a filter induced by a linear ESN with transition matrix $W_{h}$ with $\rho(W_{h})<1$. Then for any $E$ as above, for any $\epsilon>0$, and for any compact space $K\subset \mathbb{R}^{N_{in}}$, there exists $\delta>0$ such that for all $E^{\delta}$ filter induced by a transition matrix $W_{h}^{\delta}$ with $||W_{h}-W_{h}^{\delta}||<\delta$ (where $||\cdot||$ is a matrix norm), then $\|E-E^{\delta}\|_\infty=\sup_{x \in K^{\mathbb{Z}_-}} |E(x)-E^{\delta}(x)| < \epsilon$.
>
> With the lemma, we can prove the main result.
>
> **Theorem (universality of ParalESN).** (formerly corollary 4.3) ParalESN is universal in the family of fading memory filters.
>
> ---
>
> ### **W2 and Q2: Theorem 4.1**
> In this case, ESP does not hold in general for all bounded input sequences. This can be seen by taking $W_{h}=I, W_{in}=0$. In this setting, the states remain equal to the initial state, and the transient cannot be washed out. More generally, the proof can be revised without major changes as in the case $\rho<0$, and will be updated in the final version.
>
> ---
>
> ### **W3: On the hyperparameters sensitivity analysis for the deep variant**
> We have extended the sensitivity analysis to the deep variant of our model. Now, for ParalESN (deep), we graphically visualize the effect of hyperparameters such as the concatenation of the hidden states, number of layers, and the inter hyperparameters (i.e., hyperparameters for the layers after the first one). This analysis is in the updated version of the paper. Unfortunately, reporting here these results is impractical both due to character limits and due to representing violin plots in markdown.
>
> ---
>
> ### **W4 and Q3: On model selection and reported runtimes**
> We performed model selection for all models (RC and fully trainable) with a fixed time budget (24 hours maximum runtime). Reported runtimes are only for the final selected configuration, averaged across 10 different random initializations. Our goal was to showcase the proposed approach speed advantage in terms of recurrence time (Figure 2, left) and training time (Figure 4, Table 4).
> We point out that model selection is in fact a major computational burden of RC as a whole, as the models are often sensitive to hyperparameters. However, since ParalESN is significantly faster to train than standard RC models, we are able to test significantly more hyperparameters configurations for the same time budget. We can see this as an additional advantage of our approach over traditional RC: it is possible to explore the hyperparameter search space better and more efficiently.
>
> ---
>
> ### **Q4: On the Echo State Property and sensitivity to random initialization in the deep variant**
> Regarding your first point, it is fair to expect the same stability and ESP to hold even in the deep variant. As shown in [R1], the analysis is more involved in hierarchical ESNs, but we argue that, with few additional assumptions, a similar result can be recovered. Regarding your second point, we have extended our sensitivity analysis to the deep variant of our approach. This will be included in the revised version of the paper. In summary, we observe behaviour similar to that of the shallow variant. Crucial hyperparameters with a significant effect on performance remain the same even in the deep configuration (e.g., spectral radius and consequentially inter-layer spectral radius).
>
> ---
>
> [R1] C. Gallicchio and A. Micheli. “Echo state property of deep reservoir computing networks.” Cognitive Computation (2017).

---

> > ### Author Rebuttal · Reviewer_ewt1 · 2026-04-02
> >
> > Thanks for the detailed rebuttal, which completely resolves my concerns. I will maintain my positive score.

---

> > > ### Author Response · Authors · 2026-04-07
> > >
> > > Thank you very much for reading our rebuttal. We are glad that your concerns were completely resolved. We truly appreciate the constructive feedback, particularly that related to our theoretical analysis and proofs. We will certainly updated the revised version of our paper accordingly.

---

### Official Review · Reviewer_nNkg · 2026-03-09

**Soundness:** 3
**Presentation:** 3
**Significance:** 3
**Originality:** 3
**Overall Recommendation:** 4
**Confidence:** 3

**Summary:**

This paper proposes ParalESN, which adapts parallel scan techniques from State Space Models to Reservoir Computing (RC). By using diagonal linear recurrence in the complex space, it achieves O(log L) parallel processing while preserving the Echo State Property. While the theoretical bridging is clever, the experimental validation is fundamentally flawed and unconvincing.

**Compliance With Llm Reviewing Policy:**

Affirmed.

**Final Justification:**

The authors address most of my concerns. I will raise my score to 4.

**Key Questions For Authors:**

My primary concern is the severe lack of proper benchmarking. The authors borrow the core parallelization engine from SSMs, yet they only compare ParalESN against outdated RC methods. Furthermore, the datasets used (e.g., Mackey-Glass) are too simple to prove real-world expressivity.

I will raise my score if the authors provide the following in the rebuttal:

(1) Stronger Baselines: You must directly compare ParalESN against modern Deep SSMs (e.g., S4, Mamba) and linear RNNs (e.g., RWKV) in terms of both accuracy and training/inference efficiency.

(2) Large-Scale Benchmarks: You must evaluate the model on the Long Range Arena (LRA) benchmark . This is the standard for evaluating long-sequence modeling capabilities and cannot be skipped.

Without comparing against modern SSM baselines on the LRA benchmark, the practical value of this method remains unproven.

**Limitations:**

yes

**Strengths And Weaknesses:**

Strengths:

(1) Smart integration of modern SSM parallelization into traditional RC architectures.

(2) Solid mathematical proofs for preserving the Echo State Property after linearization.

(3) Successfully eliminates the strict O(L) sequential bottleneck in standard RC models.

Weaknesses:

(1) The comparison experiments are poor. Baselines are entirely outdated and ignore modern state-of-the-art linear sequence models.

(2) The evaluation relies on simple datasets and lacks complex, real-world benchmarks such as Long Range Arena benchmark.

(3) The hidden hardware costs of complex-domain arithmetic are not properly addressed.

---

> ### Author Rebuttal · Authors · 2026-03-30
>
> Thank you very much for taking the time to provide valuable feedback. We are glad that the Reviewer recognized the smart integration of modern SSM parallelization into RC architectures.
>
> ---
>
> ### **W1 and Q1: On benchmarks and baselines**
> The primary aim of our work is to advance research in RC, and our choice of baselines and datasets reflects this goal. While we agree that some tasks (e.g., Mackey-Glass) may not demonstrate real-world expressivity, our forecasting benchmarks include complex real-world time series such as ETTh1/2 and ETTm1/2 from [R1]. We believe that the superiority of ParalESN over traditional RC in the ETT tasks does provide meaningful insights into its real-world expressivity.
>
> Regarding the baselines, we do compare our approach against modern deep SSMS such as Mamba and LRU, see Table 4. We agree that S4 is missing and extended the experiments accordingly. Below, we summarize S4 results and note that ParalESN still achieves competitive performance while being more efficient.
>
> **(sMNIST)**
> | Model | Params | Accuracy | Time (min.) | Emissions (kg) | Energy (kWh) |
> |---|---|---|---|---|---|
> | S4 | ≈160k | 99.2 | 16.1† | 0.53† | 1.61† |
> | ParalESN | ≈160k | 96.2 | 2.7 | 0.01 | 0.04 |
> | ParalESN (deep) | ≈160k | 97.2 | 3.3 | 0.02 | 0.05 |
>
> **(psMNIST)**
> | Model | Params | Accuracy | Time (min.) | Emissions (kg) | Energy (kWh) |
> |---|---|---|---|---|---|
> | S4 | ≈160k | 97.9 | 9.87† | 0.35† | 1.07† |
> | ParalESN | ≈160k | 96.9 | 2.8 | 0.01 | 0.04 |
> | ParalESN (deep) | ≈160k | 95.2 | 3.1 | 0.02 | 0.05 |
>
> † We expect the real training time to be higher. The GPU used at submission time was no longer available and we had to use a different, more powerful GPU.
>
> ---
>
> ### **W2 and Q2: On Long Range Arena benchmarks**
> Our goal is broader than evaluating long-range propagation in isolation: ParalESN positions itself as a scalable, parallelizable general-purpose RC architecture. For these reasons, we evaluate it across memory, forecasting, and time-series classification.
>
> We agree that LRA represents a valuable benchmark for long-range sequence modeling. We updated the paper with evaluations on LRA benchmarks, including Image, ListOps, Text, and PathFinder.  Due to the relatively short rebuttal window and scale of these tasks, it was not feasible to include other LRA tasks (e.g., PathX). For the same reasons, it was not feasible to run our own training for the reported baselines and track metrics such as training time or energy consumption. However, the computational advantage of our approach is extensively demonstrated in other experiments.
>
> Our approach outperforms a wide variety of transformer-based models. While not able to consistently reach the same performance of S4/S5, LRU, and RWKV, ParalESN (deep) achieves competitive performance despite leveraging untrained reservoir dynamics and without requiring backpropagation-through-time.
>
> **(Long Range Arena)**
> | Model | ↑ Image | ↑ ListOps | ↑ Text | ↑ PathFinder | ↑ Avg |
> |:---|:---:|:---:|:---:|:---:|:---:|
> | Random | 10.00 | 10.00 | 50.00 | 50.00 | 30.00 |
> | *(Transformer-based)* | | | | | |
> | Transformer | 42.4 | 36.4 | 64.3 | 71.4 | 53.6 |
> | Reformer | 38.1 | 37.3 | 56.1 | 68.5 | 50.0 |
> | BigBird | 40.8 | 36.1 | 64.0 | 74.9 | 53.9 |
> | Linear Trans. | 42.3 | 16.1 | 65.9 | 75.3 | 49.9 |
> | Performer | 42.8 | 18.0 | 65.4 | 77.1 | 50.8 |
> | *(SSM-based)* | | | | | |
> | S4 | 88.7 | 59.6 | 86.8 | 94.2 | 82.3 |
> | S5 | 88.0 | 62.2 | 89.3 | 95.3 | 84.7 |
> | LRU | - | 60.2 | 89.4 | 95.1 | 81.6 |
> | *(RNN-based)* | | | | | |
> | RWKV | 70.5 | 55.9 | 86.0 | 58.4 | 67.7 |
> | *(Ours)* | | | | | |
> | ParalESN (deep) | 57.2 | 39.6 | 80.2 | 77.7 | 63.7 |
>
> ---
>
> ### **W3: On the hidden hardware costs of complex-domain arithmetic**
> We already account for the hidden hardware costs of complex-domain arithmetic. When measuring and reporting metrics such as training time (Figure 4), energy consumption, and carbon emissions (Table 4), these costs are implicitly captured, as these metrics reflect the computational overhead of complex-domain arithmetic executed on current hardware. Overall, our conclusions on the efficiency of ParalESN are grounded in actual training time, energy consumption, and carbon emissions.
>
> Additionally, in PyTorch (which we use in our experiments), complex-valued arithmetic is generally not yet as optimized as its real-valued counterpart. Moreover, complex tensors are still a beta feature in PyTorch. This means that our experiments are, in a sense, conducted under conservative conditions for our method: with more optimized libraries for complex arithmetic, ParalESN could potentially be even more efficient than the traditional real-valued RC methods we use for comparison.
>
> We agree this was not explicit enough, and we will clarify this in the revision.
>
> ---
>
> [R1] H. Zhou et al. “Informer: Beyond efficient transformer for long sequence time-series forecasting.” AAAI conference on artificial intelligence (2021).

---

> > ### Author Rebuttal · Reviewer_nNkg · 2026-04-03
> >
> > Thanks for the detailed response. It address most of my concerns. So I will raise my score to 4.

---

> > > ### Author Response · Authors · 2026-04-07
> > >
> > > Thank you very much for reading our rebuttal and for raising your score. We are glad that our rebuttal adequately addresses your concerns. We truly appreciate the reviewer efforts in suggesting additional baselines and benchmarks, which we will certainly include in the revised version of our paper.

---

### Official Review · Reviewer_2cHv · 2026-03-13

**Soundness:** 4
**Presentation:** 3
**Significance:** 2
**Originality:** 2
**Overall Recommendation:** 4
**Confidence:** 3

**Summary:**

This paper proposes ParalESN (Parallel Echo State Network), an improved Reservoir Computing (RC) architecture. The core problem is that traditional Echo State Networks face two bottlenecks:
1. time series must be processed sequentially step by step
2. the memory overhead of high-dimensional reservoirs is prohibitive due to the dense N_h × N_h transition matrix.

ParalESN draws inspiration from State Space Models (SSMs) and the Linear Recurrent Unit (LRU), replacing the traditional dense nonlinear recurrence with a diagonal linear recurrence in the complex space. The diagonal structure brings two benefits:
1. the recurrence can be parallelized via associative scan, reducing time complexity from O(T) to O(log T)
2. the transition matrix only needs to store N_h diagonal elements instead of N_h² parameters
Following the diagonal recurrence, a 1-D convolutional mixing layer is added to introduce nonlinearity and cross-dimension interaction, while the entire reservoir remains untrained. only the final linear readout layer is trained.

On the theoretical side, the paper proves three results:
1. the necessary and sufficient condition for ParalESN to satisfy the Echo State Property is that all diagonal elements have modulus less than 1
2. any linear reservoir can be equivalently represented in complex diagonal form
3. ParalESN with an MLP readout is universal in approximating fading memory filters, matching the expressive power of traditional ESNs.

Experiments cover time series regression (memory tasks, chaotic forecasting, ETT series) and classification (UEA/UCR, sMNIST/psMNIST). Results show that ParalESN matches or outperforms traditional ESNs in accuracy while training roughly an order of magnitude faster. On sMNIST/psMNIST, ParalESN approaches the accuracy of fully trainable models such as LSTMs, Transformers, LRU, and Mamba, while reducing computational cost and energy consumption by orders of magnitude. The deep variant, ParalESN (deep), ranks highest on most tasks.

Overall, this work bridges the gap between traditional Reservoir Computing and modern efficient sequence modeling in deep learning, enabling RC to process data in parallel and scale to larger dimensions while preserving the training advantage of not requiring backpropagation.

**Compliance With Llm Reviewing Policy:**

Affirmed.

**Final Justification:**

addresses all concerns

**Key Questions For Authors:**

I do not have any major questions. If I had to raise one, it would be whether the parallelization and scalability advantages of ESN can be validated on truly industrial-scale data. For instance, all the datasets used in this paper are small-scale. From a practical application perspective, one might question whether this approach is genuinely useful or simply another case of the academic community benchmarking within its own echo chamber. However, this is not really the authors' fault. The entire ESN/RC research community has traditionally used these benchmarks for comparison, and the datasets in this paper are standard benchmarks that have been used in the RC community for a long time. So my concern is directed more at the ESN field as a whole rather than at the authors specifically.

For other specific questions about the paper itself, please refer to the weaknesses discussed above.

**Limitations:**

yes

**Strengths And Weaknesses:**

I have already expressed most of my views in the summary. Let me briefly add a few more points:
1. Soundness: The three proofs are sound.
2. Presentation: I am not entirely sure whether the results in this paper can be perfectly reproduced, but after understanding the underlying principles, the implementation seems fairly straightforward. So I do not consider this a major concern.
3. Significance: Moderate. The tasks that ESNs can handle could certainly be done better with large-scale data and LLMs, but for applications such as physical system and chaotic time series prediction, edge devices, and online learning with fast adaptation, an approach with extremely low training cost and deployment overhead can indeed be valuable.
4. Originality: Moderate. The core idea is quite simple, but the complete theoretical analysis built around it does demonstrate originality. I honestly do not understand why this criterion is separated from soundness. if a theoretical paper is sound, it is naturally original. However, since the two are treated as distinct criteria, I can only note that the simplicity of the idea warrants a slight deduction.

The weaknesses I want to highlight do not fit neatly into the categories above, so I will state them separately:
1. The reservoir is randomly initialized and then fixed, which means it may not yield an optimal feature representation for a specific task. Although the paper proves universality in theory, how well a finite-dimensional random reservoir can approximate the target function in practice is highly dependent on hyperparameter choices. The hyperparameter sensitivity analysis in Appendix G confirms this: performance degrades significantly when the spectral radius, phase, or leaky rate are not properly tuned.
2. The paper claims that ParalESN is as expressive as any linear ESN, but this equivalence argument relies on diagonalization. If a particular dense-matrix ESN happens to exploit near-non-diagonalizable structure to achieve special dynamical behavior, ParalESN may not be able to precisely replicate that behavior under finite numerical precision. In other words, there may be a gap between the theoretical expressiveness equivalence and actual performance, but the paper does not discuss how large this gap might be or on what tasks it could manifest. To be clear, this is not an issue that would cause ParalESN to break down, but it is a subtle crack between the theoretical conclusions and practical behavior.
3. The mixing layer uses a 1-D convolution with fixed random weights, where the kernel size k is a hyperparameter that needs to be tuned. This design imposes a locality prior on the cross-dimension interaction pattern. For tasks that require global interactions across dimensions, this could become a bottleneck.

---

> ### Author Rebuttal · Authors · 2026-03-30
>
> Thank you very much for your positive evaluation of our paper. We are glad you recognize this work bridges the gap between reservoir computing (RC) and modern sequence modeling.
>
> ---
>
> ### **W1: On reservoir initialization and hyperparameters**
> As the reviewer correctly points out, the choice of hyperparameters is crucial for approximating the target function and, consequently, for performance. However, rather than a weakness of our approach, this is an inherent aspect of RC. Leveraging randomly weighted reservoirs means that their initialization is critical for producing rich untrained dynamics.
>
> While this implies that model selection is particularly important for ParalESN, the same holds true for traditional ESNs. However, since ParalESN is considerably faster to train (thanks to parallelization), for the same time budget, we are able to test significantly more hyperparameter configurations. This also mitigates the fact that our approach has a relatively richer hyperparameter search space compared to traditional shallow and deep ESNs. Moreover, linear recurrence is generally more interpretable, and it can help with understanding the impact of reservoir design choices such as spectral radius and input scaling. One such example is the input mixing of the $W_{in}$ matrix, which is set to the optimal value $\sqrt{1-|\lambda|^2}$, as derived in [R1].
>
>
> ---
>
> ### **W2: On the expressivity of ParalESN**
> We address this point in three parts.
>
> From a theoretical perspective. As stated in Proposition 4.2, the equivalence holds "with probability 1" precisely because the set of non-diagonalizable (defective) matrices has Lebesgue measure zero in the space of all square matrices. Since ESN recurrent matrices are randomly initialized with i.i.d. entries drawn from a continuous distribution, the probability of sampling a defective matrix is exactly zero. Hence, the theoretical equivalence holds almost surely for any randomly initialized ESN.
> We will add in the revision a revised proposition, which clarifies the claim.
>
> Regarding near-non-diagonalizable matrices. Near-defective matrices may have ill-conditioned eigenvectors matrices $V$, making the change of basis $V Λ V^{-1}$ numerically sensitive. However, we emphasize that this concern would only be relevant if ParalESN needed to numerically diagonalize a given dense matrix at runtime, which it does not. ParalESN directly parameterizes the recurrence with a diagonal matrix; no change-of-basis computation is ever performed. The diagonalization argument is used solely in the proof of Proposition 4.2 to establish that the function classes are equivalent, not as an algorithmic step. Therefore, the numerical conditioning of $V$ does not affect ParalESN's actual computation.
>
> Finally, in practice, we note that, to the best of our knowledge, there is no known class of tasks where a defective or near-defective recurrent matrix structure is necessary or beneficial. Moreover, under the Echo State Property ($|λ_i| < 1$), any transient dynamics arising from near-defective structure (e.g., polynomial growth from Jordan blocks) are exponentially damped, limiting their practical impact. Our empirical results support this: ParalESN matches or outperforms traditional ESNs (Tables 2, 3, 4) and is, overall, the top-performing RC model (Figure 3, left). Thus, empirical evidence suggests there is no practical gap between the two model classes.
>
> ---
>
> ### **W3: On the mixing layer and capturing global interactions**
> Let us point out that the choice of employing a 1-D convolution in the mixing layer is motivated by the goal of reducing its memory footprint, making it easier to build larger reservoirs.
>
> Regarding the limitation of the 1-D convolution for tasks requiring global interactions across dimensions. Two straightforward approaches can be applied easily to counterbalance this potential limitation. (i) One approach would be to set the kernel size equal to the hidden size of the states produced by the reservoir, i.e., $k = N_{h}$. In this case, the mixing layer would still be significantly more memory-efficient than a dense matrix, as we would only need to store $N_{h}$ additional weights since the same kernel is reused across all timesteps. (ii) Another approach is to enlarge the global cross-dimension interaction by stacking multiple ParalESN blocks (i.e., ParalESN (deep)), thereby increasing the effective receptive field across dimensions.
>
> ---
>
> ### **On the reproducibility of our results**
> Despite the straightforward implementation of our approach, upon acceptance, we plan to release our code to ease reproducibility and future works.
>
> ---
>
> [R1] A. Orvieto et al. “Resurrecting recurrent neural networks for long sequences.” International Conference on Machine Learning (2023).

---

> > ### Author Rebuttal · Reviewer_2cHv · 2026-04-03
> >
> > I kept all comments and ratings.

---

> > > ### Author Response · Authors · 2026-04-07
> > >
> > > Thank you very much for reading our rebuttal. We are glad that your concerns were adequately addressed. We truly appreciate the time and effort invested in evaluating our manuscript.

---

### Decision · Program_Chairs · 2026-04-30

**Decision:**

Accept (regular)

**Comment:**

This paper introduces the ParalESN (Parallel Echo State Network) model, which modernizes the reservoir computing framework with ideas from modern linear recurrent models such as state-space models and the LRU. Empirical results are shown on time series benchmarks and pixel-level classification tasks, showing competitive empirical results with increased computational efficiency due to the reservoir computing framework of not requiring trainable dynamics. After rebuttal, all reviewers are in favor for acceptance. Remaining concerns include scope of empirical results and realistic impact; for example, noticeable performance gaps start showing on more difficult tasks such as Long Range Arena. Overall, the paper is a solid contribution and I recommend acceptance.